# Memory-Efficient Approximation Algorithms for MAX-ᴋ-Cᴜᴛ and Correlation Clustering

**Nimita Shinde**
IITB-Monash Research Academy, Mumbai, India
nimitas@iitb.ac.in

**Vishnu Narayanan**
Industrial Engineering and Operations Research
IIT Bombay, Mumbai, India
vishnu@iitb.ac.in

**James Saunderson**
Electrical and Computer Systems Engineering
Monash University, Clayton, Australia
james.saunderson@monash.edu

## Abstract

MAX-ᴋ-Cᴜᴛ and correlation clustering are fundamental graph partitioning problems. For a graph $G = (V, E)$ with $n$ vertices, the methods with the best approximation guarantees for MAX-ᴋ-Cᴜᴛ and the MAX-Aɢʀᴇᴇ variant of correlation clustering involve solving SDPs with $\mathcal{O}(n^2)$ constraints and variables. Large-scale instances of SDPs, thus, present a memory bottleneck. In this paper, we develop simple polynomial-time Gaussian sampling-based algorithms for these two problems that use $\mathcal{O}(n + |E|)$ memory and nearly achieve the best existing approximation guarantees. For dense graphs arriving in a stream, we eliminate the dependence on $|E|$ in the storage complexity at the cost of a slightly worse approximation ratio by combining our approach with sparsification.

## 1 Introduction

Semidefinite programs (SDPs) arise naturally as a relaxation of a variety of problems such as $k$-means clustering [5], correlation clustering [6] and MAX-ᴋ-Cᴜᴛ [14]. In each case, the decision variable is an $n \times n$ matrix and there are $d = \Omega(n^2)$ constraints. While reducing the memory bottleneck for large-scale SDPs has been studied quite extensively in literature [9, 11, 19, 36], all these methods use memory that scales linearly with the number of constraints and also depends on either the rank of the optimal solution or an approximation parameter. A recent Gaussian-sampling based technique to generate a near-optimal, near-feasible solution to SDPs with smooth objective function involves replacing the decision variable $X$ with a zero-mean random vector whose covariance is $X$ [27]. This method uses at most $\mathcal{O}(n + d)$ memory, independent of the rank of the optimal solution. However, for SDPs with $d = \Omega(n^2)$ constraints, these algorithms still use $\Omega(n^2)$ memory and provide no advantage in storage reduction. In this paper, we show how to adapt the Gaussian sampling-based approach of [27] to generate an approximate solution with provable approximation guarantees to MAX-ᴋ-Cᴜᴛ, and to the MAX-Aɢʀᴇᴇ variant of correlation clustering on a graph $G = (V, E)$ with arbitrary edge weights using only $\mathcal{O}(|V| + |E|)$ memory.

35th Conference on Neural Information Processing Systems (NeurIPS 2021).

## 1.1 MAX-K-CUT

MAX-K-CUT is the problem of partitioning the vertices of a weighted undirected graph $G = (V, E)$ into $k$ distinct parts, such that the total weight of the edges across the parts is maximized. If $w_{ij}$ is the edge weight corresponding to edge $(i, j) \in E$, then the cut value of a partition is $\texttt{CUT} = \sum_{i \text{ and } j \text{ are in different partitions}} w_{ij}$. Consider the standard SDP relaxation of MAX-K-CUT

$$\max_{X \succeq 0} \quad \langle C, X \rangle \quad \text{subject to} \quad \begin{cases} \text{diag}(X) = \mathbb{1} \\ X_{ij} \geq -\frac{1}{k-1} \quad i \neq j, \end{cases} \qquad \text{(k-Cut-SDP)}$$

where $C = \frac{k-1}{2k} L_G$ is a scaled Laplacian. Frieze and Jerrum [14] developed a randomized rounding scheme that takes an optimal solution $X^\star$ of (k-Cut-SDP) and produces a random partitioning satisfying

$$\mathbb{E}[\texttt{CUT}] = \sum_{ij \in E, i < j} w_{ij} \text{Pr}(i \text{ and } j \text{ are in different partitions}) \geq \alpha_k \langle C, X^\star \rangle \geq \alpha_k \text{opt}_k^G, \qquad (1)$$

where $\text{opt}_k^G$ is the optimal $k$-cut value and $\alpha_k = \min_{-1/(k-1) \leq \rho \leq 1} \frac{kp(\rho)}{(k-1)(1-\rho)}$, where $p(\rho)$ is the probability that $i$ and $j$ are in different partitions given that $X_{ij} = \rho$. The rounding scheme proposed in [14], referred to as the FJ rounding scheme in the rest of the paper, generates $k$ i.i.d. samples, $z_1, \ldots, z_k \sim \mathcal{N}(0, X^\star)$ and assigns vertex $i$ to part $p$, if $[z_p]_i \geq [z_l]_i$ for all $l = 1, \ldots, k$.

## 1.2 Correlation clustering

In correlation clustering, we are given a set of $|V|$ vertices together with the information indicating whether pairs of vertices are similar or dissimilar, modeled by the edges in the sets $E^+$ and $E^-$ respectively. The MAX-AGREE variant of correlation clustering seeks to maximize

$$\mathcal{C} = \sum_{ij \in E^-} w_{ij}^- \mathbb{1}_{[i,j \text{ in different clusters}]} + \sum_{ij \in E^+} w_{ij}^+ \mathbb{1}_{[i,j \text{ in the same cluster}]}.$$

Define $G^+ = (V, E^+)$ and $G^- = (V, E^-)$. A natural SDP relaxation of MAX-AGREE [6] is

$$\max_{X \succeq 0} \quad \langle C, X \rangle \quad \text{subject to} \quad \begin{cases} \text{diag}(X) = \mathbb{1} \\ X_{ij} \geq 0 \quad i \neq j, \end{cases} \qquad \text{(MA-SDP)}$$

where $C = L_{G^-} + W^+$, $L_{G^-}$ is the Laplacian of the graph $G^-$ and $W^+$ is the weighted adjacency matrix of the graph $G^+$. Charikar et al. [10] (see also Swamy [30]) propose a rounding scheme that takes an optimal solution $X_G^\star$ of (MA-SDP) and produces a random clustering $\mathcal{C}$ satisfying

$$\mathbb{E}[\mathcal{C}] \geq 0.766 \langle C, X_G^\star \rangle \geq 0.766 \text{opt}_{CC}^G, \qquad (2)$$

where $\text{opt}_{CC}^G$ is the optimal clustering value. The rounding scheme proposed in [10], referred to as the CGW rounding scheme in the rest of the paper, generates either $k = 2$ or $k = 3$ i.i.d. zero-mean Gaussian samples with covariance $X_G^\star$ and uses them to define $2^k$ clusters.

## 1.3 Contributions

We now summarize key contributions of the paper.

**Gaussian sampling for MAX-K-CUT.** Applying Gaussian sampling-based Frank-Wolfe given in [27] directly to (k-Cut-SDP) uses $n^2$ memory. We, however, show how to extend the approach from [27] to MAX-K-CUT by proposing an alternate SDP relaxation for the problem, and combining it with the FJ rounding scheme to generate a $k$-cut with nearly the same approximation guarantees as stated in (1) (see Proposition 1) using $\mathcal{O}(n + |E|)$ memory. A key highlight of our approach is that while the approximation ratio remains close to the state-of-the-art result in (1), reducing it by a factor of $1 - 5\epsilon$ for $\epsilon \in (0, 1/5)$, the memory used is independent of $\epsilon$. We summarize our result as follows.

**Proposition 1.** *For $\epsilon \in (0, 1/5)$, our $\mathcal{O}\left(\frac{n^{2.5}|E|^{1.25}}{\epsilon^{2.5}} \log(n/\epsilon) \log(|E|)\right)$-time method outlined in Section 3 uses $\mathcal{O}(n + |E|)$ memory and generates a $k$-cut for the graph $G = (V, E)$ whose expected value satisfies $\mathbb{E}[\texttt{CUT}] \geq \alpha_k (1 - 5\epsilon) \text{opt}_k^G$, where $\text{opt}_k^G$ is the optimal $k$-cut value.*

**Gaussian sampling for MAX-AGREE.** The structure of (MA-SDP) is similar to (k-Cut-SDP), however, the cost matrix in (MA-SDP) is no longer PSD or diagonally dominant, a property that plays an important role in our analysis in the case of MAX-K-CUT. Despite this, we show how to generate a $(1 - 7\epsilon)0.766$-optimal clustering using $\mathcal{O}(n + |E|)$ memory. Our approach makes a small sacrifice in the approximation ratio (as compared to (2)), however, the memory used remains independent of $\epsilon$.

**Proposition 2.** *For $\epsilon \in (0, 1/7)$, our $\mathcal{O}\left(\frac{n^{2.5}|E|^{1.25}}{\epsilon^{2.5}} \log(n/\epsilon) \log(|E|)\right)$-time method outlined in Section 4 uses $\mathcal{O}(n + |E|)$ memory and generates a clustering of graph $G = (V, E)$ whose expected value satisfies $\mathbb{E}[\mathcal{C}] \geq 0.766(1 - 7\epsilon)\mathrm{opt}_{CC}^G$, where $\mathrm{opt}_{CC}^G$ is the optimal clustering value.*

The constructive proof outline of Propositions 1 and 2 is given in Sections 3 and 4 respectively.

**Memory reduction using graph sparsification.** Propositions 1 and 2 state that the memory used by our approach is $\mathcal{O}(n + |E|)$. However, for dense graphs, the memory used by our method becomes $\Theta(n^2)$. In this setting, to reduce the memory used, we first need to change the way we access the problem instance. We assume that the input (weighted) graph $G$ arrives edge-by-edge, eliminating the need to store the entire dense graph. We then replace it with a $\tau$-spectrally close graph $\tilde{G}$ (see Definition 1) with $\mathcal{O}(n \log n/\tau^2)$ edges. Next, we compute an approximate solution to the new problem defined on the sparse graph using $\mathcal{O}(n \log n/\tau^2)$ memory. For MAX-K-CUT and MAX-AGREE, we show that this method generates a solution with provable approximation guarantees.

## 1.4 Literature review

We first review key low memory algorithms for linearly constrained SDPs.

Burer and Monteiro [9] proposed a nonlinear programming approach which replaces the PSD decision variable with its low-rank factorization in SDPs with $d$ linear constraints. If the selected value of rank $r$ satisfies $r(r + 1) \geq 2d$ and the constraint set is a smooth manifold, then any second-order critical point of the nonconvex problem is a global optimum [8]. Another approach, that requires $\Theta(d + nr)$ working memory, is to first determine (approximately) the subspace in which the (low) rank-$r$ solution to an SDP lies and then solve the problem over the (low) $r$-dimensional subspace [11].

Alternatively, randomized sketching to a low dimensional subspace is often used as a low-memory alternative to storing a matrix decision variable [31, 34]. Recently, such sketched variables have been used to generate a low-rank approximation of a near-optimal solution to SDPs [37]. The working memory required to compute a near-optimal solution and generate its rank-$r$ approximation using the algorithmic framework proposed by Yurtsever et al. [37] is $\mathcal{O}(d + rn/\zeta)$ for some sketching parameter $\zeta \in (0, 1)$. Gaussian sampling-based Frank-Wolfe [27] uses $\mathcal{O}(n + d)$ memory to generate zero-mean Gaussian samples whose covariance represents a near-optimal solution to the SDPs with $d$ linear constraints. This eliminates the dependency on the rank of the near-optimal solution or the accuracy to which its low rank approximation is computed.

However, the two problems considered in this paper have SDP relaxations with $n^2$ constraints, for which applying the existing low-memory techniques provide no benefit since the memory requirement of these techniques depends on the number of constraints in the problem. These problems have been studied extensively in literature as we see below.

**MAX-K-CUT.** MAX-K-CUT and its dual MIN-K-PARTITION have applications in frequency allocation [12] and generating lower bound on co-channel interference in cellular networks [7]. These problems have been studied extensively in the literature [20, 26, 29]. The SDP-based rounding scheme given in [14] has also been adapted for similar problems of capacitated MAX-K-CUT [16] and approximate graph coloring [21]. In each case, however, the SDP relaxation has $\Omega(n^2)$ constraints. Alternative heuristic methods have been proposed in [13, 17, 23, 25], however, these methods generate a feasible cut which only provides a lower bound on the optimal cut value.

**Correlation clustering.** Charikar et al. [10], Swamy [30] provide 0.766-approximation schemes for MAX-AGREE each of which involve solving (MA-SDP). For large-scale applications, data streaming techniques have been studied quite extensively for various clustering problems, such as $k$-means and $k$-median [3, 24]. Ahn et al. [1] propose a single-pass, $\tilde{\mathcal{O}}(|E| + n\epsilon^{-10})$-time $0.766(1 - \epsilon)$-approximation algorithm for MAX-AGREE that uses $\tilde{\mathcal{O}}(n/\epsilon^2)$ memory. In contrast, to achieve the same approxima-

tion guarantee, our approach uses $\mathcal{O}\left(n + \min\left\{|E|, \frac{n\log n}{\epsilon^2}\right\}\right)$ memory which is equal to $\mathcal{O}(n+|E|)$ for sparse graphs, and is independent of $\epsilon$. Furthermore, the computational complexity of our approach has a better dependence on $\epsilon$ given by $\mathcal{O}\left(\frac{n^{2.5}}{\epsilon^{2.5}}\min\{|E|, \frac{n\log n}{\epsilon^2}\}^{1.25}\log(n/\epsilon)\log(|E|)\right)$ which is at most $\mathcal{O}\left(\frac{n^{3.75}}{\epsilon^5}(\log n)^{1.25}\log(n/\epsilon)\log(|E|)\right)$. Moreover, our approach is algorithmically simple to implement.

## 1.5  Outline

In Section 2, we review the Gaussian sampling-based Frank-Wolfe method [27] to compute a near-feasible, near-optimal solution to SDPs with linear equality and inequality constraints. In Sections 3 and 4 respectively, we adapt the Gaussian sampling-based approach to give an approximation algorithm for MAX-k-CUT and MAX-AGREE respectively, that use only $\mathcal{O}(n + |E|)$ memory, proving Propositions 1 and 2 respectively. In Section 5, we show how to combine our methods with streaming spectral sparsification to reduce the memory required to $\mathcal{O}(n\log n/\epsilon^2)$ for dense graphs presented edge-by-edge in a stream. We provide some preliminary computational results for MAX-AGREE in Section 6, and conclude our work and discuss possible future directions in Section 7. All proofs are deferred to the Supplementary material.

**Notations.**  The matrix inner product is denoted by $\langle A, B\rangle = \text{Tr}\left(A^T B\right)$. The vector of diagonal entries of a matrix $X$ is $\text{diag}(X)$, and $\text{diag}^*(x)$ is a diagonal matrix with the vector $x$ on the diagonal. The notations $\mathcal{O}, \Omega, \Theta$ have the usual complexity interpretation and $\tilde{\mathcal{O}}$ suppresses the dependence on $\log n$. An undirected edge $(i,j)$ in the set $E$ is denoted using $(i,j) \in E$ and $ij \in E$ interchangably.

## 2  Gaussian Sampling-based Frank-Wolfe

Consider a smooth, concave function $g$ and define the trace constrained SDP

$$\max_{X\in\mathcal{S}} \quad g(\mathcal{B}(X)), \qquad\qquad \text{(BoundedSDP)}$$

where $\mathcal{S} = \{\text{Tr}(X) \le \alpha, X \succeq 0\}$ and $\mathcal{B}(\cdot) : \mathbb{S}^n \to \mathbb{R}^d$ is a linear mapping that projects the variable from $\binom{n+1}{2}$-dimensional space to a $d$-dimensional space. One algorithmic approach to solving (BoundedSDP) is to use the Frank-Wolfe algorithm [18] which, in this case, computes an $\epsilon$-optimal solution by taking steps of the form $X_{t+1} = (1 - \gamma_t)X_t + \gamma_t \alpha h_t h_t^T$, where $\gamma_t \in [0, 1]$ and unit vectors $h_t$'s arise from approximately solving a symmetric eigenvalue problem that depends only on $\mathcal{B}(X_t)$ and $g(\cdot)$. Standard convergence results show that an $\epsilon$-optimal solution is reached after $\mathcal{O}(C_g^u/\epsilon)$ iterations, where $C_g^u$ is an upper bound on the curvature constant of $g$ [18].

**Frank-Wolfe with Gaussian sampling.**  The Gaussian sampling technique of [27] replaces the matrix-valued iterates, $X_t$, with Gaussian random vectors $z_t \sim \mathcal{N}(0, X_t)$. The update, at the level of samples, is then $z_{t+1} = \sqrt{1-\gamma_t}z_t + \sqrt{\gamma_t\alpha}\,\zeta_t h_t$, where $\zeta_t \sim \mathcal{N}(0, 1)$. Note that $z_{t+1}$ is also a zero-mean Gaussian random vector with covariance equal to $X_{t+1} = (1 - \gamma_t)X_t + \gamma_t\alpha h_t h_t^T$. Furthermore, to track the change in the objective function value, it is sufficient to track the value $v_t = \mathcal{B}(X_t)$, and compute $v_{t+1}$ such that $v_{t+1} = (1 - \gamma_t)v_t + \gamma_t\mathcal{B}(\alpha h_t h_t^T)$. Thus, computing the updates to the decision variable and tracking the objective function value only requires the knowledge of $z_t \sim \mathcal{N}(0, X_t)$ and $\mathcal{B}(X_t)$, which can be updated without explicitly storing $X_t$, thereby reducing the memory used.

Algorithm 1 [27] describes, in detail, Frank-Wolfe algorithm with Gaussian sampling when applied to (BoundedSDP). It uses at most $\mathcal{O}(n + d)$ memory at each iteration, and after at most $\mathcal{O}(C_g^u/\epsilon)$ iterations, returns a sample $\widehat{z}_\epsilon \sim \mathcal{N}(0, \widehat{X}_\epsilon)$, where $\widehat{X}_\epsilon$ is an $\epsilon$-optimal solution to (BoundedSDP).

### 2.1  SDP with linear equality and inequality constraints

Consider an SDP with linear objective function and a bounded feasible region,

$$\max_{X\succeq 0} \quad \langle C, X\rangle \quad \text{subject to} \quad \left\{ \begin{array}{l} \mathcal{A}^{(1)}(X) = b^{(1)} \\ \mathcal{A}^{(2)}(X) \ge b^{(2)}, \end{array} \right. \qquad \text{(SDP)}$$

**Algorithm 1:** (`FWGaussian`) Frank-Wolfe Algorithm with Gaussian Sampling [27]

**Input** : Input data for (BoundedSDP), stopping criteria $\epsilon$, accuracy parameter $\eta$, upper bound $C_g^u$ on the curvature constant, probability $p$ for the subproblem `LMO`

**Output** : $z \sim \mathcal{N}(0, \widehat{X}_\epsilon)$ and $v = \mathcal{B}(\widehat{X}_\epsilon)$, where $\widehat{X}_\epsilon$ is an $\epsilon$-optimal solution of (BoundedSDP)

1 **Function** `FWGaussian`:
2     Select initial point $X_0 \in \mathcal{S}$; set $v_0 \leftarrow \mathcal{B}(X_0)$ and sample $z_0 \sim \mathcal{N}(0, X_0)$
3     $t \leftarrow 0, \gamma \leftarrow 2/(t+2)$
4     $(h_t, q_t) \leftarrow \texttt{LMO}(\mathcal{B}^*(\nabla g(v_t)), \frac{1}{2}\eta\gamma C_g^u)$
5     **while** $\langle q_t - v_t, \nabla g(v_t) \rangle > \epsilon$ **do**
6        $(z_{t+1}, v_{t+1}) \leftarrow \texttt{UpdateVariable}(z_t, v_t, h_t, q_t, \gamma)$
7        $t \leftarrow t+1, \gamma \leftarrow 2/(t+2)$
8        $(h_t, q_t) \leftarrow \texttt{LMO}(\mathcal{B}^*(\nabla g(v_t)), \frac{1}{2}\eta\gamma C_g^u, p)$
9     **end**
10 **return** $(z_t, v_t)$
11 **Function** `LMO`$(J, \delta)$:
12     Find a unit vector $h$ such that with probability at least $1 - p$,
     $\alpha\lambda = \alpha\langle hh^T, J \rangle \geq \max_{d \in \mathcal{S}} \alpha\langle d, J \rangle - \delta$
13     **if** $\lambda \geq 0$ **then** $q \leftarrow \mathcal{B}(\alpha hh^T)$
14     **else** $q \leftarrow 0, h \leftarrow 0$
15 **return** $(h, q)$
16 **Function** `UpdateVariable`$(z, v, h, q, \gamma)$:
17     $z \leftarrow (\sqrt{1-\gamma})z + \sqrt{\gamma\alpha}\, h\zeta$ where $\zeta \sim \mathcal{N}(0, 1)$
18     $v \leftarrow (1-\gamma)v + \gamma q$
19 **return** $(z, v)$

where $\mathcal{A}^{(1)}(\cdot) : \mathbb{S}_+^n \to \mathbb{R}^{d_1}$ and $\mathcal{A}^{(2)}(\cdot) : \mathbb{S}_+^n \to \mathbb{R}^{d_2}$ are linear maps. To use Algorithm 1, the linear constraints are penalized using a smooth penalty function. Let $u_l = \langle A_l^{(1)}, X \rangle - b_l^{(1)}$ for $l = 1, \dots, d_1$ and $v_l = b_l^{(2)} - \langle A_l^{(2)}, X \rangle$ for $l = 1, \dots, d_2$. For $M > 0$, the smooth function $\phi_M(\cdot) : \mathbb{R}^{d_1+d_2} \to \mathbb{R}$,

$$\phi_M(u, v) = \frac{1}{M} \log \left( \sum_{i=1}^{d_1} e^{M(u_i)} + \sum_{i=1}^{d_1} e^{M(-u_i)} + \sum_{j=1}^{d_2} e^{M(v_j)} \right), \quad \text{satisfies} \qquad (3)$$

$$\max\left\{ \|u\|_\infty, \max_i v_i \right\} \leq \phi_M(u, v) \leq \frac{\log(2d_1 + d_2)}{M} + \max\left\{ \|u\|_\infty, \max_i v_i \right\}.$$

We add this penalty term to the objective of (SDP) and define

$$\max_{X \succeq 0} \left\{ \langle C, X \rangle - \beta\phi_M(\mathcal{A}^{(1)}(X) - b^{(1)}, b^{(2)} - \mathcal{A}^{(2)}(X)) \; : \; \text{Tr}(X) \leq \alpha \right\}, \qquad \text{(SDP-LSE)}$$

where $\alpha, \beta$ and $M$ are appropriately chosen parameters. Algorithm 1 then generates a Gaussian sample with covariance $\widehat{X}_\epsilon$ which is an $\epsilon$-optimal solution to (SDP-LSE). It is also a near-optimal, near-feasible solution to (SDP). This result is a slight modification of [27, Lemma 3.2] which only provides bounds for SDPs with linear equality constraints.

**Lemma 1.** *For $\epsilon > 0$, let $(X^\star, \vartheta^\star, \mu^\star)$ be an optimal primal-dual solution to* (SDP) *and its dual, and let $\widehat{X}_\epsilon$ be an $\epsilon$-optimal solution to* (SDP-LSE). *If $\beta > \|\vartheta^\star\|_1 + \|\mu^\star\|_1$ and $M > 0$, then*

$$\langle C, X^\star \rangle - \frac{\beta\log(2d_1 + d_2)}{M} - \epsilon \leq \langle C, \widehat{X}_\epsilon \rangle \leq \langle C, X^\star \rangle + (\|\vartheta^\star\|_1 + \|\mu^\star\|_1)\frac{\beta\frac{\log(2d_1+d_2)}{M} + \epsilon}{\beta - \|\vartheta^\star\|_1 - \|\mu^\star\|_1},$$

$$\max\left\{ \|\mathcal{A}^{(1)}(X) - b^{(1)}\|_\infty, \max_i \left( b_i^{(2)} - \mathcal{A}_i^{(2)}(X) \right) \right\} \leq \frac{\beta\frac{\log(2d_1+d_2)}{M} + \epsilon}{\beta - \|\vartheta^\star\|_1 - \|\mu^\star\|_1}.$$

## 3 Application of Gaussian Sampling to (k-Cut-SDP)

In this section, we look at the application of Gaussian sampling to MAX-$k$-CUT. Since Algorithm 1 uses $\mathcal{O}(n^2)$ memory when solving (k-Cut-SDP), we define a new SDP relaxation of MAX-$k$-CUT with the same approximation guarantee, but with $\mathcal{O}(|E|)$ constraints. We then apply Algorithm 1 to this new relaxation, and show how to round the solution to achieve nearly the same approximation ratio as given in (1). Let

$$\alpha_k = \min_{-1/(k-1) \leq \rho \leq 1} \frac{kp(\rho)}{(k-1)(1-\rho)}, \tag{4}$$

where $p(X_{ij})$ is the probability that vertices $i$ and $j$ are in different partitions. If $X$ is feasible for (k-Cut-SDP) and CUT is the value of the $k$-cut generated by the FJ rounding scheme, then

$$\mathbb{E}[\text{CUT}] = \sum_{ij \in E, i<j} w_{ij} p(X_{ij})$$

$$\geq \sum_{ij \in E, i<j} \frac{k-1}{k} w_{ij}(1 - X_{ij})\alpha_k = \alpha_k \langle C, X \rangle. \tag{5}$$

Frieze and Jerrum [14] derive a lower bound on $\alpha_k$, showing that the method gives a nontrivial approximation guarantee. Observe that (5) depends only on the values $X_{ij}$ if $(i,j) \in E$.

**A new SDP relaxation of MAX-$k$-CUT.** We relax the constraints in (k-Cut-SDP) to define

$$\max_{X \succeq 0} \quad \langle C, X \rangle \quad \text{subject to} \quad \begin{cases} \text{diag}(X) = \mathbb{1} \\ X_{ij} \geq -\frac{1}{k-1} \quad (i,j) \in E, i < j. \end{cases} \tag{k-Cut-Rel}$$

Since (k-Cut-Rel) is a relaxation of (k-Cut-SDP), its optimal objective function value provides an upper bound on $\langle C, X^\star \rangle$, where $X^\star$ is an optimal solution to (k-Cut-SDP), and hence, on the optimal $k$-cut value $\text{opt}_k^G$. Note that the bound in (5) holds true even if we replace $X^\star$ by an optimal solution to (k-Cut-Rel) since it depends on the value of $X_{ij}$ only if $(i,j) \in E$. Furthermore, when the FJ rounding scheme is applied to the solution of (k-Cut-Rel), it satisfies the approximation guarantee on the expected value of the generated $k$-cut given in (1), i.e., $\mathbb{E}[\text{CUT}] \geq \alpha_k \text{opt}_k^G$.

**Using Algorithm 1.** We now have an SDP relaxation of MAX-$k$-CUT that has $n + |E|$ constraints. Penalizing the linear constraints in (k-Cut-Rel) using the function $\phi_M(\cdot)$ (3), Algorithm 1 can now be used to generate $k$ samples with covariance $\widehat{X}_\epsilon$ which is an $\epsilon$-optimal solution to

$$\max_{X \succeq 0} \left\{ \langle C, X \rangle - \beta \phi_M \left( \text{diag}(X) - \mathbb{1}, -\frac{1}{k-1} - e_i^T X e_j \right) : (i,j) \in E, \text{Tr}(X) \leq n \right\}. \tag{k-Cut-LSE}$$

**Optimality and feasibility results for (k-Cut-Rel).** Given an $\epsilon$-optimal solution to (k-Cut-LSE), we show that it is also a near-optimal, near-feasible solution to (k-Cut-Rel).

**Lemma 2.** *For $\epsilon \in (0, 1/2)$, let $X_R^\star$ be an optimal solution to* (k-Cut-Rel) *and let $\widehat{X}_\epsilon$ be an $\epsilon \text{Tr}(C)$-optimal solution to* (k-Cut-LSE). *For $\beta = 6\text{Tr}(C)$ and $M = 6 \frac{\log(2n+|E|)}{\epsilon}$, we have*

$$(1 - 2\epsilon)\langle C, X_R^\star \rangle \leq \langle C, \widehat{X}_\epsilon \rangle \leq (1 + 4\epsilon)\langle C, X_R^\star \rangle \quad \text{and} \tag{6}$$

$$\|\text{diag}(\widehat{X}_\epsilon) - \mathbb{1}\|_\infty \leq \epsilon, \quad [\widehat{X}_\epsilon]_{ij} \geq -\frac{1}{k-1} - \epsilon, \quad (i,j) \in E, i < j. \tag{7}$$

**Generating a feasible solution to MAX-$k$-CUT.** Since $\widehat{X}_\epsilon$ might not necessarily be feasible to (k-Cut-Rel), we cannot apply the FJ rounding scheme to the samples $z_i \sim \mathcal{N}(0, \widehat{X}_\epsilon)$. We, therefore, generate samples $z_i^f \sim \mathcal{N}(0, X^f)$ using the procedure given in Algorithm 2 such that $X^f$ is a feasible solution to (k-Cut-Rel) and $\langle C, X^f \rangle$ is close to $\langle C, \widehat{X}_\epsilon \rangle$.

We can now apply the FJ rounding scheme to $z_1^f, \ldots, z_k^f$ as given in Lemma 3.

**Lemma 3.** *For $G = (V, E)$, let $\text{opt}_k^G$ be the optimal $k$-cut value and let $X_R^\star$ be an optimal solution to* (k-Cut-Rel). *For $\epsilon \in \left(0, \frac{1}{4}\right)$, let $\widehat{X}_\epsilon \succeq 0$ satisfy (6) and (7). Let $z_1^f, \ldots, z_k^f$ be random vectors generated by Algorithm 2 with input $z_i, \ldots, z_k \sim \mathcal{N}(0, \widehat{X}_\epsilon)$ and let CUT denote the value of a $k$-cut generated by applying the FJ rounding scheme to $z_1^f, \ldots, z_k^f$. For $\alpha_k$ as defined by (4), we have*

$$\alpha_k(1 - 4\epsilon)\text{opt}_k^G \leq \alpha_k(1 - 4\epsilon)\langle C, X_R^\star \rangle \leq \mathbb{E}[\text{CUT}] \leq \text{opt}_k^G. \tag{8}$$

---

**Algorithm 2:** Generate Gaussian samples with covariance feasible to (k-Cut-Rel)

---

**Input** : Sample $z_i \sim \mathcal{N}(0, \widehat{X}_\epsilon)$ for $i = 1, \ldots, k$ and $\text{diag}(\widehat{X}_\epsilon)$

**Output** : $z_i^f \sim \mathcal{N}(0, X^f)$ for $i = 1, \ldots, k$ with $X^f$ feasible to (k-Cut-Rel)

---

**1 Function** GeneratekSamples:

**2**    **for** $i = 1, \ldots, k$ **do**

**3**        Set err $= \max\{0, \max_{(i,j) \in E, i<j} -1/(k-1) - [\widehat{X}_\epsilon]_{ij}\}$

**4**        Set $\overline{z}_i = z_i + \sqrt{\text{err}}\, y\, \mathbb{1}$, where $y \sim \mathcal{N}(0,1)$

**5**        Generate $\zeta \sim \mathcal{N}\left(0, I - \text{diag}^*\left(\frac{\text{diag}(\widehat{X}_\epsilon) + \text{err}}{\max(\text{diag}(\widehat{X}_\epsilon)) + \text{err}}\right)\right)$

**6**        Set $z_i^f = \frac{\overline{z}_i}{\sqrt{\max(\text{diag}(\widehat{X}_\epsilon)) + \text{err}}} + \zeta$

**7**    **end**

**8 return** $z_1^f, \ldots, z_k^f$

---

**Computational complexity of Algorithm 1 when applied to** (k-Cut-LSE). Finally, in Lemma 4, we provide the computational complexity of the method proposed in this section, which concludes the proof of Proposition 1.

**Lemma 4.** *When the method proposed in this section (Section 3), with $p = \frac{\epsilon}{T(n,\epsilon)}$ and $T(n, \epsilon) = \frac{144 \log(2n + |E|)n^2}{\epsilon^2}$, is used to generate an approximate k-cut to* MAX-K-CUT*, the generated cut satisfies $\mathbb{E}[\mathtt{CUT}] \geq \alpha_k(1 - 5\epsilon)\text{opt}_k^G$ and runs in $\mathcal{O}\left(\frac{n^{2.5}|E|^{1.25}}{\epsilon^{2.5}} \log(n/\epsilon) \log(|E|)\right)$ time.*

## 4 Application of Gaussian Sampling to (MA-SDP)

We now look at the application of our Gaussian sampling-based method to MAX-AGREE. Algorithm 1 uses $\mathcal{O}(n^2)$ memory to generate samples whose covariance is an $\epsilon$-optimal solution to (MA-SDP). However, with the similar observation as in the case of MAX-K-CUT, we note that for any $X$ feasible to (MA-SDP), the proof of the inequality $\mathbb{E}[\mathcal{C}] \geq 0.766\langle C, X\rangle$, given in [10, Theorem 3], requires $X_{ij} \geq 0$ only if $(i, j) \in E$. We therefore, write a new relaxation of (MA-SDP),

$$\max_{X \succeq 0} \langle C, X\rangle = \langle W^+ + L_{G^-}, X\rangle \quad \text{subject to} \left\{ \begin{array}{l} X_{ii} = 1 \ \forall\, i \in \{1, \ldots, n\} \\ X_{ij} \geq 0 \ (i, j) \in E, i < j, \end{array} \right. \quad \text{(MA-Rel)}$$

with only $n + |E|$ constraints. The bound $\mathbb{E}[\mathcal{C}] \geq 0.766\langle C, X^\star\rangle \geq 0.766\text{opt}_{CC}^G$ on the expected value of the clustering holds even if the clustering is generated by applying the CGW rounding scheme to an optimal solution $X^\star$ of (MA-Rel). To use Algorithm 1, we penalize the constraints in (MA-Rel) and define

$$\max_{X \succeq 0} \left\{\langle C, X\rangle - \beta\phi_M(\text{diag}(X) - \mathbb{1}, -e_i^T X e_j) : (i, j) \in E, \text{Tr}(X) \leq n\right\}. \quad \text{(MA-LSE)}$$

**Optimality and feasibility results for** (MA-Rel). Algorithm 1 is now used to generate $z \sim \mathcal{N}(0, \widehat{X}_\epsilon)$, where $\widehat{X}_\epsilon$ is an $\epsilon$-optimal solution to (MA-LSE). We show in Lemma 5 that $\widehat{X}_\epsilon$ is also a near-optimal, near-feasible solution to (MA-Rel).

**Lemma 5.** *For $\Delta = \text{Tr}(L_{G^-}) + \sum_{ij \in E^+} w_{ij}^+$, $\epsilon \in \left(0, \frac{1}{4}\right)$, let $X_G^\star$ be an optimal solution to (MA-Rel) and $\widehat{X}_\epsilon$ be an $\epsilon\Delta$-optimal solution to (MA-LSE). Setting $\beta = 4\Delta$ and $M = 4\frac{\log(2n + |E|)}{\epsilon}$, we have*

$$(1 - 4\epsilon)\langle C, X_G^\star\rangle \leq \langle C, \widehat{X}_\epsilon\rangle \leq (1 + 4\epsilon)\langle C, X_G^\star\rangle \quad \text{and} \tag{9}$$

$$\|\text{diag}(\widehat{X}_\epsilon) - \mathbb{1}\|_\infty \leq \epsilon, \quad [\widehat{X}_\epsilon]_{ij} \geq -\epsilon, \quad (i, j) \in E, i < j. \tag{10}$$

**Generating an approximate clustering.** The CGW rounding scheme can only be applied if we have a feasible solution to (MA-Rel). We, therefore, use a modified version of Algorithm 2, with Step 3 replaced by err $= \max\{0, \max_{(i,j) \in E, i<j} -[\widehat{X}_\epsilon]_{ij}\}$ and input $z_1, z_2, z_3 \sim \mathcal{N}(0, \widehat{X}_\epsilon)$, to generate zero-mean Gaussian samples whose covariance is a feasible solution to (MA-Rel). Finally, we apply the CGW rounding scheme to the output of the modified of Algorithm 2.

**Lemma 6.** *Let $X_G^\star$ be an optimal solution to* (MA-Rel)*. For $\epsilon \in (0, 1/6)$, let $\widehat{X}_\epsilon \succeq 0$ satisfy* (9) *and* (10)*, and let $z_1^f, z_2^f, z_3^f$ be random vectors generated by Algorithm 2 with input $z_1, z_2, z_3 \sim \mathcal{N}(0, \widehat{X}_\epsilon)$. Let $\mathrm{opt}_{CC}^G$ denote the optimal clustering value for the graph $G = (V, E)$ and let $\mathcal{C}$ denote the value of the clustering generated from the random vectors $z_1^f, z_2^f, z_3^f$ using the CGW rounding scheme. Then*

$$\mathbb{E}[\mathcal{C}] \geq 0.766(1 - 6\epsilon)\langle C, X_G^\star \rangle \geq 0.766(1 - 6\epsilon)\mathrm{opt}_{CC}^G. \tag{11}$$

**Computational complexity of Algorithm 1 when applied to** (MA-LSE)**.**

**Lemma 7.** *When the method proposed in this section (Section 4), with $p = \frac{\epsilon}{T(n, \epsilon)}$ and $T(n, \epsilon) = \frac{64 \log(2n + |E|)n^2}{\epsilon^2}$, is used to generate an approximate clustering, the value of the clustering satisfies $\mathbb{E}[\mathcal{C}] \geq 0.766(1 - 7\epsilon)\mathrm{opt}_{CC}^G$ and runs in $\mathcal{O}\left(\frac{n^{2.5}|E|^{1.25}}{\epsilon^{2.5}} \log(n/\epsilon) \log(|E|)\right)$ time.*

This completes the proof of Proposition 2.

## 5 Sparsifying the Laplacian Cost Matrix

As seen in Sections 3 and 4, the memory requirement for generating and representing an $\epsilon$-optimal solution to (k-Cut-LSE) and (MA-LSE) is bounded by $\mathcal{O}(n + |E|)$. However, if the input graph $G$ is dense, the cost matrix will be dense and the number of inequality constraints will still be high. In this section, we consider the situation in which the dense weighted graph arrives in a stream, and we first build a sparse approximation with similar spectral properties. We refer to this additional step as *sparsifying* the cost.

**Definition 1** ($\tau$-spectral closeness)**.** *Two graphs $G$ and $\tilde{G}$ defined on the same set of vertices are said to be $\tau$-spectrally close if, for any $x \in \mathbb{R}^n$ and $\tau \in (0, 1)$,*

$$(1 - \tau)x^T L_G x \leq x^T L_{\tilde{G}} x \leq (1 + \tau)x^T L_G x. \tag{12}$$

Spectral graph sparsification has been studied quite extensively (see, e.g., [2, 15, 28]). Kyng et al. [22] propose a $\mathcal{O}(|E| \log^2 n)$-time framework to replace a dense graph $G = (V, E)$ by a sparser graph $\tilde{G} = (V, \tilde{E})$ such that $|\tilde{E}| \sim \mathcal{O}(n \log n/\tau^2)$ and $\tilde{G}$ satisfies (12) with probability $1 - \frac{1}{\mathrm{poly}(n)}$. Their algorithm assumes that the edges of the graph arrive one at a time, so that the total memory requirement is $\mathcal{O}(n \log n/\tau^2)$ rather than $\mathcal{O}(|E|)$. Furthermore, a sparse cost matrix decreases the computation time of the subproblem in Algorithm 1 since it involves matrix-vector multiplication with the gradient of the cost.

**MAX-K-CUT with sparsification.** Let $\tilde{G}$ be a sparse graph with $\mathcal{O}(n \log n/\tau^2)$ edges that is $\tau$-spectrally close to the input graph $G$. By applying the method outlined in Section 3, we can generate a $k$-cut for the graph $\tilde{G}$ (using $\mathcal{O}(n \log n/\tau^2)$ memory) whose expected value satisfies the bound (8). Note that, this generated cut is also a $k$-cut for the original graph $G$ with provable approximation guarantee as shown in Lemma 8.

**Lemma 8.** *For $\epsilon, \tau \in (0, 1/5)$, let $\widehat{X}_\epsilon$ be a near-feasible, near-optimal solution to* (k-Cut-Rel) *defined on the graph $\tilde{G}$ that satisfies* (6) *and* (7)*. Let $\mathtt{CUT}$ denote the value of the $k$-cut generated by applying Algorithm 2 followed by the FJ rounding scheme to $\widehat{X}_\epsilon$. Then the generated cut satisfies*

$$\alpha_k(1 - 4\epsilon - \tau)\mathrm{opt}_k^G \leq \mathbb{E}[\mathtt{CUT}] \leq \mathrm{opt}_k^G,$$

*where $\mathrm{opt}_k^G$ is the value of the optimal $k$-cut for the original graph $G$.*

**MAX-AGREE with sparsification.** The number of edges $|E^+|$ and $|E^-|$ in graphs $G^+$ and $G^-$ respectively determine the working memory of Algorithm 1. For dense input graphs $G^+$ and $G^-$, we sparsify them to generate graphs $\tilde{G}^+$ and $\tilde{G}^-$ with at most $\mathcal{O}(n \log n/\tau^2)$ edges and define

$$\max_{X \in \mathcal{S}} \tilde{f}(X) = \langle L_{\tilde{G}^-} + \tilde{W}^+, X \rangle - \beta\phi_M(\mathrm{diag}(X) - \mathbb{1}, -[e_i^T X e_j]_{(i,j) \in \tilde{E}}), \tag{MA-Sparse}$$

where $\mathcal{S} = \{X : \mathrm{Tr}(X) \leq n, X \succeq 0\}$, $L_{\tilde{G}^-}$ is the Laplacian of the graph $\tilde{G}^-$, $\tilde{W}^+$ is matrix with nonnegative entries denoting the weight of each edge $(i, j) \in \tilde{E}^+$, and $\tilde{E} = \tilde{E}^+ \cup \tilde{E}^-$.

Algorithm 1 then generates an $\epsilon(\text{Tr}(L_{\tilde{G}^-}) + \sum_{ij \in \tilde{E}^+} \tilde{w}_{ij}^+)$-optimal solution, $\widehat{X}_\epsilon$, to (MA-Sparse) using $\mathcal{O}(n \log n/\tau^2)$ memory. We can now use the method given in Section 4 to generate a clustering of graph $\tilde{G}$ whose expected value, $\mathbb{E}[\mathcal{C}]$, satisfies (11). The following lemma shows that $\mathcal{C}$ also represents a clustering for the original graph $G$ with provable guarantees.

**Lemma 9.** *For $\epsilon, \tau \in (0, 1/9)$, let $\widehat{X}_\epsilon$ be a near-feasible, near-optimal solution to* (MA-Sparse) *defined on the graph $\tilde{G}$ that satisfies* (9) *and* (10). *Let $\mathcal{C}$ denote the value of the clustering generated by applying Algorithm 2 followed by the CGW rounding scheme to $\widehat{X}_\epsilon$. Then, $\mathbb{E}[\mathcal{C}]$ satisfies*

$$0.766(1 - 6\epsilon - 3\tau)(1 - \tau^2)\text{opt}_{CC}^G \leq \mathbb{E}[\mathcal{C}] \leq \text{opt}_{CC}^G,$$

*where $\text{opt}_{CC}^G$ is the value of the optimal clustering of the original graph $G$.*

We summarize our results in the following lemma whose proof is given in the Supplementary material.

**Lemma 10.** *Assume that the edges of the input graph $G = (V, E)$ arrive one at a time in a stream. The procedure given in this section uses at most $\mathcal{O}(n \log n/\tau^2)$ memory and in $\mathcal{O}\left(\frac{n^{2.5}|E|^{1.25}}{\epsilon^{2.5}} \log(n/\epsilon) \log(|E|)\right)$-time, generates approximate solutions to* MAX-K-CUT *and* MAX-AGREE *that satisfy the bounds $\mathbb{E}[\mathcal{CUT}] \geq \alpha_k(1 - 5\epsilon - \tau)\text{opt}_k^G$ and $\mathbb{E}[\mathcal{C}] \geq 0.766(1 - 7\epsilon - 3\tau)(1 - \tau^2)\text{opt}_{CC}^G$ respectively.*

# 6 Computational Results

We now discuss the results of preliminary computations to cluster the vertices of a graph $G$ using the approach outlined in Section 4. The aim of numerical experiments was to verify that the bounds given in Lemma 6 were satisfied when we used the procedure outlined in Section 4 to generate a clustering for each input graph. We used the graphs from GSET dataset [35] which is a collection of randomly generated graphs. Note that the aim of correlation clustering is to generate a clustering of vertices for graphs where each edge has a label indicating 'similarity' or 'dissimilarity' of the vertices connected to that edge. We, therefore, first converted the undirected, unweighted graphs from the GSET dataset [35] into the instances of graphs with labelled edges using an adaptation of the approach used in [32, 33]. This modified approach generated a label and weight for each edge $(i, j) \in E$ indicating the amount of 'similarity' or 'dissimilarity' between vertices $i$ and $j$.

**Generating input graphs for MAX-AGREE.** In the process of label generation, we first computed the Jaccard coefficient $J_{ij} = |N(i) \cap N(j)|/|N(i) \cup N(j)|$, where $N(i)$ is the set of neighbours of $i$ for each edge $(i, j) \in E$. Next we computed the quantity $S_{ij} = \log((1 - J_{ij} + \delta)/(1 + J_{ij} - \delta))$ with $\delta = 0.05$ for each edge $(i, j) \in E$, which is a measure of the amount of 'similarity' or 'dissimilarity'. Finally, the edge $(i, j)$ was labelled as 'dissimilar' if $S_{ij} < 0$ with $w_{ij}^- = -S_{ij}$ and labelled as 'similar' with $w_{ij}^+ = S_{ij}$ otherwise.

**Experimental Setup.** We set the input parameters to $\epsilon = 0.05$, $\Delta = \text{Tr}(L_{G^-}) + \sum_{ij \in E^+} w_{ij}^+$, $\beta = 4\Delta$, $M = 4\frac{\log(2n) + |E|}{\epsilon}$. Using Algorithm 1, (MA-LSE) was solved to $\epsilon\Delta$-optimality and, we computed feasible samples using Algorithm 2. Finally, we generated two Gaussian samples and created at most four clusters by applying the 0.75-rounding scheme proposed by Swamy [30, Theorem 2.1], for simplicity. The computations were performed using MATLAB R2021a on a machine with 8GB RAM. We noted the peak memory used by the algorithm using the `profiler` command in MATLAB. The code and the GSET dataset are included in the Supplementary material.

The computational result for some randomly selected instances from the dataset is given in Table 1. We have provided the result for the rest of the graphs from GSET in the Supplementary material. First, we observed that for each input graph, the number of iterations of LMO for $\epsilon\Delta$-convergence satisfied the bound given in Proposition 1 and the infeasibility of the covariance $\widehat{X}_\epsilon$ of the generated samples was less than $\epsilon$ satisfying (10). We generated 10 pairs of i.i.d. zero-mean Gaussian samples with covariance $\widehat{X}_\epsilon$, and each in turn was used to generate a clustering for the input graph using the 0.75-rounding scheme proposed by Swamy [30]. Amongst the 10 clusterings generated for each graph, we picked the clustering with largest value denoted by $\mathcal{C}_{\text{best}}$. Note that, $\mathcal{C}_{\text{best}} \geq \mathbb{E}[\mathcal{C}] \geq 0.75(1 - 6\epsilon)\langle C, X_G^\star\rangle \geq 0.75\frac{1-6\epsilon}{1+4\epsilon}\langle C, \widehat{X}_\epsilon\rangle$, where the last inequality follows from

combining (11) with (9). Since we were able to generate the values, $\mathcal{C}_{\text{best}}$ and $\langle C, \widehat{X}_\epsilon \rangle$, we noted that the weaker bound $\mathcal{C}_{\text{best}}/\langle C, \widehat{X}_\epsilon \rangle = \text{AR} \geq 0.75(1 - 6\epsilon)/(1 + 4\epsilon)$ was satisfied by every input graph when $\epsilon = 0.05$.

Table 1 also shows the memory used by our method. Consider the dataset G1, for which the memory used by our method was $1526.35\text{kB} \approx 9.8 \times (|V| + |E^+| + |E^-|) \times 8$, where a factor of 8 denotes that MATLAB requires 8 bytes to store a real number. Similarly, we observed that our method used at most $c \times (|V| + |E^+| + |E^-|) \times 8$ memory to generate clusters for other instances from GSET, where $c \leq 33$ for every instance of the input graph, showing that the memory used was linear in the size of the input graph.

Table 1: Result of generating a clustering of graphs from GSET using the method outlined in Section 4. We have, $\text{infeas} = \max\{\|\text{diag}(X) - 1\|_\infty, \max\{0, -[\widehat{X}_\epsilon]_{ij}\}\}$, $\text{AR} = \mathcal{C}_{\text{best}}/\langle C, \widehat{X}_\epsilon \rangle$ and $0.75(1 - 6\epsilon)/(1 + 4\epsilon) = 0.4375$ for $\epsilon = 0.05$.

| Dataset | $|V|$ | $|E^+|$ | $|E^-|$ | # Iterations $(\times 10^3)$ | infeas | $\langle C, \widehat{X}_\epsilon \rangle$ | $\mathcal{C}_{\text{best}}$ | AR | Memory required (in kB) |
|---|---|---|---|---|---|---|---|---|---|
| G1 | 800 | 2453 | 16627 | 669.46 | $10^{-3}$ | 849.48 | 643 | 0.757 | 1526.35 |
| G11 | 800 | 817 | 783 | 397.2 | $6 \times 10^{-4}$ | 3000.3 | 2080 | 0.693 | 448.26 |
| G14 | 800 | 3861 | 797 | 330.02 | $8 \times 10^{-4}$ | 542.55 | 469.77 | 0.866 | 423.45 |
| G22 | 2000 | 115 | 19849 | 725.66 | $10^{-3}$ | 1792.9 | 1371.1 | 0.764 | 1655.09 |
| G32 | 2000 | 2011 | 1989 | 571.42 | $9 \times 10^{-4}$ | 7370 | 4488 | 0.609 | 1124 |
| G43 | 1000 | 248 | 9704 | 501.31 | $10^{-3}$ | 803.8 | 616.05 | 0.766 | 654.46 |
| G48 | 3000 | 0 | 6000 | 9806.22 | 0.004 | 599.64 | 461.38 | 0.769 | 736.09 |
| G51 | 1000 | 4734 | 1147 | 1038.99 | 0.001 | 676.21 | 446.29 | 0.66 | 517.09 |
| G55 | 5000 | 66 | 12432 | 2707.07 | 0.002 | 1244.2 | 901.74 | 0.724 | 1281.03 |
| G57 | 5000 | 4981 | 5019 | 574.5 | 0.005 | 18195 | 10292 | 0.565 | 812.78 |

## 7    Discussion

In this paper, we proposed a Gaussian sampling-based optimization algorithm to generate approximate solutions to MAX-K-CUT, and the MAX-AGREE variant of correlation clustering using $\mathcal{O}\left(n + \min\left\{|E|, \frac{n \log n}{\tau^2}\right\}\right)$ memory. The approximation guarantees given in [10, 14, 30] for these problems are based on solving SDP relaxations of these problems that have $n^2$ constraints. The key observation that led to the low-memory method proposed in this paper was that the approximation guarantees from literature are preserved for both problems even if we solve their weaker SDP relaxations with only $\mathcal{O}(n + |E|)$ constraints. We showed that for MAX-K-CUT, and the MAX-AGREE variant of correlation clustering, our approach nearly preserves the quality of the solution as given in [10, 14]. We also implemented the method outlined in Section 4 to generate approximate clustering for random graphs with provable guarantees. The numerical experiments showed that while the method was simple to implement, it was slow in practice. However, there is scope for improving the convergence rate of our method so that it can potentially be applied to the large-scale instances of various real-life applications of clustering.

**Extending the low-memory method to solve problems with triangle inequalities.** The known nontrivial approximation guarantees for sparsest cut problem involve solving an SDP relaxation that has $n^3$ triangle inequalities [4]. It would be interesting to see whether it is possible to simplify these SDPs in such a way that they can be combined nicely with memory efficient algorithms, and still maintain good approximation guarantees.

## Disclosure of Funding

Funding in direct support of this work: scholarship for Nimita Shinde was provided by IITB-Monash Research Academy. Additionally, Dr. James Saunderson was supported in part by an Australian Research Council Discovery Early Career Researcher Award (project number DE210101056) funded by the Australian Government.

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
