# Supplementary Material: Memory-Efficient Approximation Algorithms for MAX-K-CUT and Correlation Clustering

**Nimita Shinde**
IITB-Monash Research Academy, Mumbai, India
nimitas@iitb.ac.in

**Vishnu Narayanan**
Industrial Engineering and Operations Research
IIT Bombay, Mumbai, India
vishnu@iitb.ac.in

**James Saunderson**
Electrical and Computer Systems Engineering
Monash University, Clayton, Australia
james.saunderson@monash.edu

## A   Proofs

### A.1   Proof of Lemma 1

*Proof.* Let $\vartheta \in \mathbb{R}^{d_1}$ and $\mu \in \mathbb{R}^{d_2}$ be the dual variables corresponding to the $d_1$ equality constraints and the $d_2$ inequality constraints respectively. The dual of (SDP) is

$$\min_{\vartheta, \mu} \quad \sum_{i=1}^{d_1} b_i^{(1)} \vartheta_i + \sum_{j=1}^{d_2} b_j^{(2)} \mu_j \quad \text{subject to} \quad \left\{ \begin{array}{l} \sum_{i=1}^{d_1} \vartheta_i A_i^{(1)} + \sum_{j=1}^{d_2} A_j^{(2)} \mu_j - C \succeq 0 \\ \mu \leq 0, \end{array} \right. \quad \text{(DSDP)}$$

where $A_j^{(2)}$'s for $j = 1, \ldots, d_2$ are assumed to be symmetric.

**Lower bound on the objective.**   Let $X^\star$ be an optimal solution to (SDP) and let $X_{FW}^\star$ be an optimal solution to (SDP-LSE). For ease of notation, let

$$u = \mathcal{A}^{(1)}(X) - b^{(1)} \quad \text{and} \quad v = b^{(2)} - \mathcal{A}^{(2)}(X), \tag{1}$$

and define $(\widehat{u}_\epsilon, \widehat{v}_\epsilon)$, $(u_{FW}, v_{FW})$ and $(u^\star, v^\star)$ by substituting $\widehat{X}_\epsilon$, $X_{FW}$ and $X^\star$ respectively in (1). Since $\widehat{X}_\epsilon$ is an $\epsilon$-optimal solution to (SDP-LSE), and a feasible solution to (SDP-LSE), the following holds

$$\langle C, \widehat{X}_\epsilon \rangle - \beta \phi_M(\widehat{u}_\epsilon, \widehat{v}_\epsilon) \geq \langle C, X_{FW} \rangle - \beta \phi_M(u_{FW}, v_{FW}) - \epsilon \geq \langle C, X^\star \rangle - \beta \phi_M(u^\star, v^\star) - \epsilon. \tag{2}$$

We know that $(u^\star, v^\star)$ is feasible to (SDP), so that $\phi_M(u^\star, v^\star) \leq \frac{\log(2d_1 + d_2)}{M}$. Now, rearranging the terms, and using the upper bound on $\phi_M(u^\star, v^\star)$ and the fact that $\phi_M(\widehat{u}_\epsilon, \widehat{v}_\epsilon) \geq 0$,

$$\langle C, \widehat{X}_\epsilon \rangle \geq \langle C, X^\star \rangle - \frac{\beta \log(2d_1 + d_2)}{M} - \epsilon. \tag{3}$$

35th Conference on Neural Information Processing Systems (NeurIPS 2021).

**Upper bound on the objective.** The Lagrangian of (SDP) is $L(X, \vartheta, \mu) = \langle C, X \rangle - \sum_{i=1}^{d_1} u_i \vartheta_i + \sum_{j=1}^{d_2} v_j \mu_j$. For a primal-dual optimal pair, $(X^\star, \vartheta^\star, \mu^\star)$, and $\widehat{X}_\epsilon \succeq 0$, we have that $L(\widehat{X}_\epsilon, \vartheta^\star, \mu^\star) \leq L(X^\star, \vartheta^\star, \mu^\star)$, i.e.,

$$\langle C, \widehat{X}_\epsilon \rangle - \sum_{i=1}^{d_1} \vartheta_i^\star [\widehat{u}_\epsilon]_i + \sum_{i=j}^{d_2} \mu_j^\star [\widehat{v}_\epsilon]_j \leq \langle C, X^\star \rangle - \sum_{i=1}^{d_1} \vartheta_i^\star u_i^\star + \sum_{j=1}^{d_2} \mu_j^\star v_j^\star$$
$$\leq \langle C, X^\star \rangle.$$

Rearranging the terms, using the duality of the $\ell_1$ and $\ell_\infty$ norms, and the fact that $\mu^\star \leq 0$, gives

$$\langle C, \widehat{X}_\epsilon \rangle \leq \langle C, X^\star \rangle + \sum_{i=1}^{d_1} \vartheta_i^\star [\widehat{u}_\epsilon]_i - \sum_{j=1}^{d_2} \mu_j^\star [\widehat{v}_\epsilon]_j$$
$$\leq \langle C, X^\star \rangle + \left( \sum_{i=1}^{d_1} |\vartheta_i^\star| \right) \|\widehat{u}_\epsilon\|_\infty + \left( \sum_{j=1}^{d_2} -\mu_j^\star \right) \max_j [\widehat{v}_\epsilon]_j \tag{4}$$
$$\leq \langle C, X^\star \rangle + \|[\vartheta^\star, \mu^\star]\|_1 \max \left\{ \|\widehat{u}_\epsilon\|_\infty, \max_j [\widehat{v}_\epsilon]_j \right\}.$$

**Bound on infeasibility.** Using (4), we rewrite (2) as,

$$\beta \phi_M(\widehat{u}_\epsilon, \widehat{v}_\epsilon) \leq \langle C, \widehat{X}_\epsilon \rangle - \langle C, X^\star \rangle + \beta \phi_M(u^\star, v^\star) + \epsilon$$
$$\leq \|[\vartheta^\star, \mu^\star]\|_1 \max \left\{ \|\widehat{u}_\epsilon\|_\infty, \max_j [\widehat{v}_\epsilon]_j \right\} + \beta \frac{\log(2d_1 + d_2)}{M} + \epsilon. \tag{5}$$

Combining the lower bound on $\phi_M(\cdot)$ given in (2.1) with (5) and since $\beta > \|[\vartheta^\star, \mu^\star]\|_1$ by assumption, we have

$$\max \left\{ \|\widehat{u}_\epsilon\|_\infty, \max_j [\widehat{v}_\epsilon]_j \right\} \leq \frac{\beta \frac{\log(2d_1 + d_2)}{M} + \epsilon}{\beta - \|[\vartheta^\star, \mu^\star]\|_1}. \tag{6}$$

**Completing the upper bound on the objective.** Substituting (6) into (4) gives

$$\langle C, \widehat{X}_\epsilon \rangle \leq \langle C, X^\star \rangle + \|[\vartheta^\star, \mu^\star]\|_1 \frac{\beta \frac{\log(2d_1 + d_2)}{M} + \epsilon}{\beta - \|[\vartheta^\star, \mu^\star]\|_1}. \tag{7}$$

$\square$

## A.2 Proof of Lemma 2

*Proof.* The proof consists of three parts.

**Lower bound on the objective.** Substituting the values of $\beta$ and $M$, and replacing $\epsilon$ by $\epsilon \text{Tr}(C)$ in (3), we have

$$\langle C, \widehat{X}_\epsilon \rangle \geq \langle C, X_R^\star \rangle - 2\epsilon \text{Tr}(C). \tag{8}$$

Since the identity matrix $I$ is strictly feasible for (k-Cut-Rel), $\text{Tr}(C) \leq \langle C, X_R^\star \rangle$. Combining this fact with (8) gives,

$$\langle C, \widehat{X}_\epsilon \rangle \geq (1 - 2\epsilon) \langle C, X_R^\star \rangle.$$

**Bound on infeasibility.** For (k-Cut-Rel), let $\nu = [\nu^{(1)}, \nu^{(2)}] \in \mathbb{R}^{n+|E|}$ be a dual variable such that $\nu_i^{(1)}$ for $i = 1, \ldots, n$ are the variables corresponding to $n$ equality constraints and $\nu_{ij}^{(2)}$ for $(i, j) \in E, i < j$ are the dual variables corresponding to $|E|$ inequality constraints. Following (DSDP), the dual of (k-Cut-Rel) is

$$\min_\nu \sum_{i=1}^n \nu_i^{(1)} - \frac{1}{k-1} \sum_{\substack{ij \in E \\ i < j}} \nu_{ij}^{(2)} \quad \text{subject to} \quad \begin{cases} \text{diag}^*(\nu^{(1)}) + \sum_{\substack{ij \in E \\ i < j}} [e_i e_j^T + e_j e_i^T] \frac{\nu_{ij}^{(2)}}{2} - C \succeq 0 \\ \nu^{(2)} \leq 0. \end{cases}$$

$$\text{(Dual-Relax)}$$

Let $\nu^\star$ be an optimal dual solution. In order to bound the infeasibility using (6), we need a bound on $\|\nu^\star\|_1$ which is given by the following lemma.

**Lemma A.1.** *The value of $\|\nu^\star\|_1$ is upper bounded by $4\mathrm{Tr}(C)$.*

*Proof.* The matrix $C$ is a scaled Laplacian and so, the only off-diagonal entries that are nonzero correspond to $(i,j) \in E$ and have value less than zero. For (Dual-Relax), a feasible solution is $\nu^{(1)} = \mathrm{diag}(C)$, $\nu_{ij}^{(2)} = 2C_{ij}$ for $(i,j) \in E, i < j$. The optimal objective function value of (Dual-Relax) is then upper bounded by

$$\sum_{i=1}^{n} \nu_i^{(1)\star} - \frac{1}{k-1} \sum_{\substack{ij \in E \\ i<j}} \nu_{ij}^{(2)\star} \le \mathrm{Tr}(C) + \frac{1}{k-1}\mathrm{Tr}(C) = \frac{k}{k-1}\mathrm{Tr}(C)$$

$$\Rightarrow \quad \sum_{i=1}^{n} \nu_i^{(1)\star} \le \frac{k}{k-1}\mathrm{Tr}(C) + \frac{1}{k-1}\sum_{\substack{ij \in E \\ i<j}} \nu_{ij}^{(2)\star} \le \frac{k}{k-1}\mathrm{Tr}(C), \tag{9}$$

where the last inequality follows since $\nu^{(2)} \le 0$.

We have $\left\langle \mathrm{diag}^*(\nu^{(1)\star}) + \sum_{\substack{ij \in E \\ i<j}} [e_i e_j^T + e_j e_i^T]\frac{\nu_{ij}^{(2)\star}}{2}, \mathbb{1}\mathbb{1}^T \right\rangle - \langle C, \mathbb{1}\mathbb{1}^T \rangle \ge 0$ since both matrices are PSD. Using the fact that $\mathbb{1}$ is in the null space of $C$, we get

$$- \sum_{\substack{ij \in E \\ i<j}} \nu_{ij}^{(2)\star} \le \sum_{i=1}^{n} \nu_i^{(1)\star}. \tag{10}$$

Since $\nu^{(2)\star} \le 0$, we can write

$$\|\nu^\star\|_1 = \sum_{i=1}^{n} |\nu_i^{(1)\star}| - \sum_{\substack{ij \in E \\ i<j}} \nu_{ij}^{(2)\star} \le 2\sum_{i=1}^{n} \nu_i^{(1)\star}, \tag{11}$$

which follows from (10) and the fact that for the dual to be feasible we have $\nu^{(1)} \ge 0$ since $C$ has nonnegative entries on the diagonal. Substituting (9) in (11),

$$\|\nu^\star\|_1 \le \frac{2k}{k-1}\mathrm{Tr}(C) \le 4\mathrm{Tr}(C), \tag{12}$$

where the last inequality follows since $k/(k-1) \le 2$ for $k \ge 2$. $\qquad\square$

Since $\widehat{X}_\epsilon$ is an $\epsilon\mathrm{Tr}(C)$-optimal solution to (k-Cut-LSE), we replace $\epsilon$ be $\epsilon\mathrm{Tr}(C)$ in (6). Finally, substituting (12) into (6), and setting $\beta = 6\mathrm{Tr}(C)$ and $M = 6\frac{\log(2n+|E|)}{\epsilon}$,

$$\max\left\{ \|\mathrm{diag}(\widehat{X}_\epsilon) - \mathbb{1}\|_\infty, \max_{ij \in E, i<j} -\frac{1}{k-1} - [\widehat{X}_\epsilon]_{ij} \right\} \le \epsilon. \tag{13}$$

This condition can also be stated as

$$\|\mathrm{diag}(\widehat{X}_\epsilon) - \mathbb{1}\|_\infty \le \epsilon, \quad [\widehat{X}_\epsilon]_{ij} \ge -\frac{1}{k-1} - \epsilon \quad (i,j) \in E, i < j.$$

**Upper bound on the objective.** Substituting (13) and (12) and the values of parameters $\beta$ and $M$ into (7) gives

$$\langle C, \widehat{X}_\epsilon \rangle \le \langle C, X_R^\star \rangle + 4\mathrm{Tr}(C)\epsilon \le (1 + 4\epsilon)\langle C, X_R^\star \rangle,$$

where the last inequality follows since $\mathrm{Tr}(C) \le \langle C, X_R^\star \rangle$. $\qquad\square$

## A.3 Proof of Lemma 3

*Proof.* We first show that Algorithm 2 generates samples whose covariance is feasible to (k-Cut-Rel).

**Proposition 1.** *Given $k$ Gaussian random vectors $z_1, \ldots, z_k \sim \mathcal{N}(0, \widehat{X}_\epsilon)$, such that their covariance $\widehat{X}_\epsilon$ satisfies the inequality (7), the Gaussian random vectors $z_1^f, \ldots, z_k^f \sim \mathcal{N}(0, X^f)$ generated by Algorithm 2 have covariance $X^f$ that is a feasible solution to (k-Cut-Rel).*

*Proof.* Define $\overline{X} = \widehat{X}_\epsilon + \text{err} \mathbb{1}\mathbb{1}^T$. Note that, $\overline{X} \succeq 0$ and it satisfies the following properties:

1. Since $\widehat{X}_\epsilon$ satisfies (7), we have err $\leq \epsilon$. Combining this fact with the definition of $\overline{X}$, we have $\overline{X}_{jl} \geq -\frac{1}{k-1}$ for $(j,l) \in E, j < l$.

2. Furthermore, $\text{diag}(\overline{X}) = \text{diag}(\widehat{X}_\epsilon) + \text{err}$, which when combined with (7), gives $1 \leq \text{diag}(\overline{X}) \leq 1 + 2\text{err}$.

3. For $y \sim \mathcal{N}(0,1)$, if $\overline{z}_i = z_i + \sqrt{\text{err}} y \mathbb{1}$, i.e., it is a sum of two Gaussian random vectors, then $\overline{z}_i \sim \mathcal{N}(0, \overline{X})$.

The steps 5 and 6 of Algorithm 2 generate a zero-mean random vector $z^f$ whose covariance is

$$X^f = \frac{\overline{X}}{\max(\text{diag}(\overline{X}))} + \left( I - \text{diag}^* \left( \frac{\text{diag}(\overline{X})}{\max(\text{diag}(\overline{X}))} \right) \right), \tag{14}$$

i.e., $z^f \sim \mathcal{N}(0, X^f)$. Furthermore, $X^f$ is feasible to (k-Cut-Rel) since $\text{diag}(X^f) = \mathbb{1}$, $X_{jl}^f \geq -\frac{1}{k-1}$ for $(j,l) \in E, j < l$, and it is a sum of two PSD matrices so that $X^f \succeq 0$. $\qquad \square$

The objective function value of (k-Cut-Rel) at $X^f$ (defined in (14)) is

$$\langle C, X^f \rangle = \left\langle C, \frac{\widehat{X}_\epsilon + \text{err}\mathbb{1}\mathbb{1}^T}{\max(\text{diag}(\widehat{X}_\epsilon)) + \text{err}} + \left( I - \text{diag}^* \left( \frac{\text{diag}(\widehat{X}_\epsilon) + \text{err}}{\max(\text{diag}(\widehat{X}_\epsilon)) + \text{err}} \right) \right) \right\rangle$$

$$\underset{(i)}{\geq} \frac{\langle C, \widehat{X}_\epsilon \rangle}{\max(\text{diag}(\widehat{X}_\epsilon)) + \text{err}} \underset{(ii)}{\geq} \frac{1 - 2\epsilon}{1 + 2\epsilon} \langle C, X_R^\star \rangle \underset{(iii)}{\geq} (1 - 4\epsilon) \langle C, X_R^\star \rangle, \tag{15}$$

where (i) follows from the fact that both $C$ and $\frac{\text{err}\mathbb{1}\mathbb{1}^T}{\max(\text{diag}(\widehat{X}_\epsilon)) + \text{err}} + I - \text{diag}^* \left( \frac{\text{diag}(\widehat{X}_\epsilon) + \text{err}}{\max(\text{diag}(\widehat{X}_\epsilon)) + \text{err}} \right)$ are PSD and so, their inner product is nonnegative, (ii) follows from Lemma 2 and the fact that err $\leq \epsilon$, and (iii) uses the fact that $1 - 2\epsilon \geq (1 + 2\epsilon)(1 - 4\epsilon)$. Let $\mathbb{E}[\text{CUT}]$ denote the value of the cut generated from the samples $z_i^f$'s. Combining (15) with the inequality $\frac{\mathbb{E}[\text{CUT}]}{\langle C, X^f \rangle} \geq \alpha_k$ (see (5)), we have

$$\mathbb{E}[\text{CUT}] \geq \alpha_k \langle C, X^f \rangle \geq \alpha_k (1 - 4\epsilon) \langle C, X_R^\star \rangle \geq \alpha_k (1 - 4\epsilon) \text{opt}_k^G. \tag{16}$$

$\qquad \square$

## A.4 Proof of Lemma 4

*Proof.* We use Algorithm 1 with $p = \frac{\epsilon}{T(n,\epsilon)}$ and $T(n,\epsilon) = \frac{144 \log(2n + |E|)n^2}{\epsilon^2}$ to generate an $\epsilon \text{Tr}(C)$-optimal solution to (k-Cut-LSE). We first bound the outer iteration complexity, i.e., the number of iterations of Algorithm 1 until convergence. This value also denotes the number of times the subproblem LMO is solved.

**Upper bound on outer iteration complexity.** Let the objective function of (k-Cut-LSE) be

$$g(X) = \langle C, X \rangle - \beta \phi_M \left( \text{diag}(X) - \mathbb{1}, \left[ -\frac{1}{k-1} - e_i^T X e_j \right]_{(i,j) \in E} \right).$$

**Theorem 1.** *Let $g(X)$ be a concave and differentiable function and $X^\star$ an optimal solution of (k-Cut-LSE). Let $C_g^u$ be an upper bound on the curvature constant of $g$, and let $\eta \geq 0$ be the accuracy parameter for LMO, then, $X_t$ satisfies*

$$-g(X_t) + g(X^\star) \leq \frac{2C_g^u(1+\eta)}{t+2},$$

*with probability at least $(1-p)^t \geq 1 - tp$.*

The result follows from [1, Theorem 1] when LMO generates a solution with approximation error at most $\frac{1}{2}\eta\gamma C_g^u$ with probability $1 - p$. Now, $\eta \in (0,1)$ is an appropriately chosen constant, and from [4, Lemma 3.1], an upper bound $C_f^u$ on the curvature constant of $g(X)$ is $\beta M n^2$. Thus, after at most

$$T = \frac{2C_g^u(1+\eta)}{\epsilon\mathrm{Tr}(C)} - 2 = \frac{2\beta M n^2(1+\eta)}{\epsilon\mathrm{Tr}(C)} - 2 \tag{17}$$

iterations, Algorithm 1 generates an $\epsilon\mathrm{Tr}(C)$-optimal solution to (k-Cut-LSE).

**Bound on the approximate $k$-cut value.** From Theorem 1, we see that after at most $T$ iterations, Algorithm 1 generates a solution $\widehat{X}_\epsilon$ that satisfies the bounds in Lemma 2 with probability with at least $1 - \epsilon$ when $p = \frac{\epsilon}{T(n,\epsilon)}$. Consequently, the bound given in (15) also holds with probability at least $1 - \epsilon$. And so, the expected value of $\langle C, X^f \rangle$ is $\mathbb{E}[\langle C, X^f \rangle] \geq (1-4\epsilon)\langle C, X_R^\star \rangle(1-\epsilon) \geq (1-5\epsilon)\langle C, X_R^\star \rangle$. Finally, from (16), the expected value of the $k$-cut, denoted by $\mathbb{E}[\mathtt{CUT}]$, is bounded as

$$\mathbb{E}[\mathtt{CUT}] = \mathbb{E}_L[\mathbb{E}_G[\mathtt{CUT}]] \geq \alpha_k\mathbb{E}_L[\langle C, X^f \rangle] \geq \alpha_k(1-5\epsilon)\langle C, X_R^\star \rangle \geq \alpha_k(1-5\epsilon)\mathrm{opt}_k^G,$$

where $\mathbb{E}_L[\cdot]$ denotes the expectation over the randomness in the subproblem LMO and $\mathbb{E}_G[\cdot]$ denotes the expectation over random Gaussian samples.

Finally, we compute an upper bound on the complexity of each iteration, i.e., inner iteration complexity, of Algorithm 1.

**Upper bound on inner iteration complexity.** At each iteration $t$, Algorithm 1 solves the subproblem LMO, which generates a unit vector $h$, such that

$$\alpha\langle hh^T, \nabla g_t \rangle \geq \max_{d \in \mathcal{S}} \alpha\langle d, \nabla g_t \rangle - \frac{1}{2}\eta\gamma_t C_g^u, \tag{18}$$

where $\gamma_t = \frac{2}{t+2}$, $\nabla g_t = \nabla g(X_t)$ and $\mathcal{S} = \{X \succeq 0 : \mathrm{Tr}(X) \leq n\}$. Note that this problem is equivalent to approximately computing maximum eigenvector of the matrix $\nabla g_t$ which can be done using Lanczos algorithm [2].

**Lemma A.2** (Convergence of Lanczos algorithm). *Let $\rho \in (0,1]$ and $p \in (0,1/2]$. For $\nabla g_t \in \mathbb{S}_n$, the Lanczos method [2], computes a vector $h \in \mathbb{R}^n$, that satisfies*

$$h^T \nabla g_t h \geq \lambda_{\max}(\nabla g_t) - \frac{\rho}{8}\|\nabla g_t\| \tag{19}$$

*with probability at least $1 - 2p$, after at most $q \geq \frac{1}{2} + \frac{1}{\sqrt{\rho}}\log(n/p^2)$ iterations.*

This result is an adaptation of [2, Theorem 4.2] which provides convergence of Lanczos to approximately compute minimum eigenvalue and the corresponding eigenvector of a symmetric matrix. Let $N = \frac{1}{2} + \frac{1}{\sqrt{\rho}}\log(n/p^2)$. We now derive an upper bound on $N$.

Comparing (19) and (18), we see that

$$\frac{1}{2}\eta\gamma_t C_g^u = \alpha\frac{\rho}{8}\|\nabla g_t\|$$

$$\Rightarrow \frac{1}{\rho} = \frac{\alpha\|\nabla g_t\|}{4\eta\gamma_t C_g^u}$$

Substituting the value of $\gamma_t$ in the equation above, and noting that $\gamma_t = \frac{2}{t+2} \geq \frac{2}{T+2}$, we have

$$\frac{1}{\rho} = \frac{\alpha\|\nabla g_t\|(t+2)}{8\eta C_g^u} \leq \frac{\alpha\|\nabla g_t\|(T+2)}{8\eta C_g^u} = \frac{\alpha\|\nabla g_t\|(1+\eta)}{4\eta\epsilon\mathrm{Tr}(C)}, \tag{20}$$

where the last equality follows from substituting the value of $T$ (see (17)). We now derive an upper bound on $\|\nabla g_t\|$.

**Lemma A.3.** *Let* $g(X) = \langle C, X \rangle - \beta \phi_M \left( \mathrm{diag}(X) - \mathbb{1}, \left[ -\frac{1}{k-1} - e_i^T X e_j \right]_{(i,j) \in E} \right)$, *where* $\phi_M(\cdot)$ *is defined in* (3). *We have* $\|\nabla g_t\| \leq \mathrm{Tr}(C)(1 + 6(\sqrt{2|E| + n}))$.

*Proof.* For the function $g(X)$ as defined in the lemma, $\nabla g_t = C - \beta D$, where $D$ is matrix such that $D_{ii} \in [-1, 1]$ for $i = 1, \ldots, n$, $D_{ij} \in [-1, 1]$ for $(i, j) \in E$, and $D_{ij} = 0$ for $(i, j) \notin E$. Thus, we have

$$\max_k |\lambda_k(D)| \leq \sqrt{\mathrm{Tr}(D^T D)} = \sqrt{\sum_{i,j=1}^n |D_{ij}|^2} \leq \sqrt{2|E| + n}, \tag{21}$$

where the last inequality follows since there are at most $2|E|$ off-diagonal and $n$ diagonal nonzero entries in the matrix $D$ with each nonzero entry in the range $[-1, 1]$. Now,

$$
\begin{aligned}
\|\nabla g_t\| = \|C - \beta D\| &\underset{(i)}{\leq} \|C\| + \| - \beta D\| \\
&\leq \max_i |\lambda_i(C)| + \max_i |\lambda_i(-\beta D)| \\
&\underset{(ii)}{\leq} \mathrm{Tr}(C) + \beta \sqrt{2|E| + n} \\
&\underset{(iii)}{\leq} \mathrm{Tr}(C)(1 + 6(\sqrt{2|E| + n})).
\end{aligned}
$$

where (i) follows from the triangle inequality for the spectral norm of $C - \beta D$, (ii) follows from (21) and since $C$ is graph Laplacian and a positive semidefinite matrix, and (iii) follows by substituting $\beta = 6\mathrm{Tr}(C)$ as given in Lemma 2. $\qquad \square$

Substituting $\alpha = n$, and the bound on $\|\nabla g_t\|$ in (20), we have

$$\frac{1}{\rho} \leq \frac{1 + \eta}{4\eta} \frac{n(1 + 6(\sqrt{2|E| + n}))}{\epsilon}, \quad \text{and}$$

$$N = \frac{1}{2} + \frac{1}{\sqrt{\rho}} \log(n/p^2) \leq \frac{1}{2} + \sqrt{\frac{1 + \eta}{4\eta}} \sqrt{\frac{n(1 + 6(\sqrt{2|E| + n}))}{\epsilon}} \log(n/p^2) = N^u.$$

Finally, each iteration of Lanczos method performs a matrix-vector multiplication with $\nabla g_t$, which has at most $2|E| + n$ number of nonzero iterations, and $\mathcal{O}(n)$ additional arithmetic operations. Thus, the computational complexity of Lanczos method is $\mathcal{O}(N^u(|E| + n))$. Moreover, Algorithm 1 performs $\mathcal{O}(|E| + n)$ additional arithmetic operations so that the total inner iteration complexity is $\mathcal{O}(N^u(|E| + n))$, which can be written as $\mathcal{O}\left( \frac{\sqrt{n}|E|^{1.25}}{\sqrt{\epsilon}} \log(n/p^2) \right)$.

**Computational complexity of Algorithm 1.** Now, substituting $p = \frac{\epsilon}{T(n,\epsilon)}$, we have

$$\log\left(\frac{n}{p^2}\right) = \log\left(\frac{(144)^2 n^5 (\log(2n + |E|))^2}{\epsilon^6}\right) \leq \log\left(\frac{(5.3n)^6}{\epsilon^6}\right) = 6\log\left(\frac{5.3n}{\epsilon}\right),$$

where the inequality follows since $|E| \leq \binom{n-1}{2}$, $\left(\log\left(2n + \binom{n-1}{2}\right)\right)^2 \leq n$ for $n \geq 1$ and $(5.3)^6 \geq (144)^2$. Substituting the upper bound on $\log(n/p^2)$ in $N^u$, and combining the inner iteration complexity, $\mathcal{O}(N^u(|E| + n))$, and outer iteration complexity, $T$, we get a $\mathcal{O}\left( \frac{n^{2.5}|E|^{1.25}}{\epsilon^{2.5}} \log(n/\epsilon) \log(|E|) \right)$-time algorithm. $\qquad \square$

## A.5 Proof of Lemma 5

*Proof.* We need to prove four inequalities given in Lemma 5.

**Lower bound on the objective, $\langle C, \widehat{X}_\epsilon \rangle$.** Substituting the values of $\beta$ and $M$, and replacing $\epsilon$ by $\epsilon \mathrm{Tr}(C)$ in (3), we have

$$\langle C, \widehat{X}_\epsilon \rangle \geq \langle C, X_G^\star \rangle - 2\epsilon \left( \mathrm{Tr}(L_{G^-}) + \sum_{ij \in E^+} w_{ij}^+ \right). \tag{22}$$

Since $0.5I + 0.5\mathbb{1}\mathbb{1}^T$ is feasible for (MA-Rel), $0.5(\mathrm{Tr}(L_{G^-}) + \sum_{ij \in E^+} w_{ij}^+) \leq \langle C, X_G^\star \rangle$. Combining this fact with (22), we have

$$\langle C, \widehat{X}_\epsilon \rangle \geq (1 - 4\epsilon)\langle C, X_G^\star \rangle.$$

**Bound on infeasibility.** Let $E = E^- \cup E^+$ and let $\nu = [\nu^{(1)}, \nu^{(2)}] \in \mathbb{R}^{n+|E|}$ be the dual variable such that $\nu^{(1)}$ is the dual variable corresponding to the $n$ equality constraints and $\nu^{(2)}$ is the dual variable for $|E|$ inequality constraints. Following (DSDP), the dual of (MA-Rel) is

$$\min_\nu \quad \sum_{i=1}^n \nu_i^{(1)} \quad \text{subject to} \quad \begin{cases} \mathrm{diag}^*(\nu^{(1)}) + \sum_{\substack{ij \in E \\ i<j}} [e_i e_j^T + e_j e_i^T] \frac{\nu_{ij}^{(2)}}{2} - C \succeq 0 \\ \nu^{(2)} \leq 0, \end{cases} \tag{Dual-CC}$$

where $C = L_{G^-} + W^+$. Let $\nu^\star$ be an optimal dual solution. We derive an upper bound on $\|\nu^\star\|_1$ in the following lemma, which is then used to bound the infeasibility using (6).

**Lemma A.4.** *The value of $\|\nu^\star\|_1$ is upper bounded by $2 \left( \mathrm{Tr}(L_{G^-}) + \sum_{ij \in E^+} w_{ij}^+ \right)$.*

*Proof.* For (Dual-CC), $\nu_i^{(1)} = [L_{G^-}]_{ii} + \sum_{j:ij \in E^+} w_{ij}^+$ for $i = 1, \ldots, n$, and $\nu_{ij}^{(2)} = 2[L_{G^-}]_{ij}$ for $(i, j) \in E, i < j$ is a feasible solution. The optimal objective function value of (Dual-CC) is then upper bounded as

$$\sum_{i=1}^n \nu_i^{(1)\star} \leq \mathrm{Tr}(L_{G^-}) + \sum_{ij \in E^+} w_{ij}^+. \tag{23}$$

We have $\left\langle \mathrm{diag}^*(\nu^{(1)\star}) + \sum_{\substack{ij \in E \\ i<j}} [e_i e_j^T + e_j e_i^T] \frac{\nu_{ij}^{(2)\star}}{2} - C, \mathbb{1}\mathbb{1}^T \right\rangle \geq 0$ since both matrices are PSD.

Using the fact that $\langle L_{G^-}, \mathbb{1}\mathbb{1}^T \rangle = 0$, and rearranging the terms, we have

$$-\sum_{\substack{ij \in E \\ i<j}} \nu_{ij}^{(2)\star} \leq \sum_{i=1}^n \nu_i^{(1)\star} - \sum_{ij \in E^+} w_{ij}^+.$$

Since $\nu^{(2)\star} \leq 0$, we can write

$$\|\nu^\star\|_1 = \sum_{i=1}^n |\nu_i^{(1)\star}| - \sum_{\substack{ij \in E \\ i<j}} \nu_{ij}^{(2)\star} \leq 2\sum_{i=1}^n \nu_i^{(1)\star} - \sum_{ij \in E^+} w_{ij}^+, \tag{24}$$

where we have used the fact that for any dual feasible solution, $\nu_i^{(1)} \geq [L_{G^-}]_{ii} \geq 0$ for all $i = 1, \ldots, n$. Substituting (23) in (24),

$$\|\nu^\star\|_1 \leq 2\mathrm{Tr}(L_{G^-}) + \sum_{ij \in E^+} w_{ij}^+ \leq 2 \left( \mathrm{Tr}(L_{G^-}) + \sum_{ij \in E^+} w_{ij}^+ \right). \tag{25}$$

$\square$

For $\Delta = \text{Tr}(L_{G^-}) + \sum_{ij \in E^+} w_{ij}^+$, $\widehat{X}_\epsilon$ is an $\epsilon\Delta$-optimal solution to (MA-LSE). And so, we replace $\epsilon$ be $\epsilon\Delta$ in (6). Now, substituting (25) and the values of $\beta$ and $M$ into (6), we get

$$\max \left\{ \|\text{diag}(\widehat{X}_\epsilon) - \mathbb{1}\|_\infty, \ \max_{ij \in E, i < j} -[\widehat{X}_\epsilon]_{ij} \right\} \le \epsilon. \tag{26}$$

This condition can also be stated as

$$\|\text{diag}(\widehat{X}_\epsilon) - \mathbb{1}\|_\infty \le \epsilon, \quad [\widehat{X}_\epsilon]_{ij} \ge -\epsilon \quad (i,j) \in E, i < j.$$

Substituting (26), (25) and the values of the parameters $\beta$ and $M$ into (7) gives

$$\langle C, \widehat{X}_\epsilon \rangle \le \langle C, X_G^\star \rangle + 2 \left( \text{Tr}(L_{G^-}) + \sum_{ij \in E^+} w_{ij}^+ \right) \epsilon \le (1 + 4\epsilon)\langle C, X_G^\star \rangle,$$

where the last inequality follows since $I$ is a feasible solution to (MA-Rel). $\qquad\square$

## A.6 Proof of Lemma 6

*Proof.* We first note that Algorithm 2 generates a samples whose covariance is feasible to (MA-Rel).

**Proposition 2.** *Let $z_1, z_2 \sim \mathcal{N}(0, \widehat{X}_\epsilon)$ be Gaussian random vectors such that their covariance $\widehat{X}_\epsilon$ satisfies the inequality* (10). *Replace Step 3 of Algorithm 2 with* $\text{err} = \max\{0, \max_{(i,j) \in E, i < j} -[\widehat{X}_\epsilon]_{ij}\}$. *The Gaussian random vectors $z_1^f, z_2^f \sim \mathcal{N}(0, X^f)$ generated by the modified Algorithm 2 have covariance that is feasible to* (MA-Rel).

The proof of Proposition 2 is the same as the proof of Proposition 1. Now, let

$$X^f = \frac{\widehat{X}_\epsilon + \text{err}\mathbb{1}\mathbb{1}^T}{\max(\text{diag}(\widehat{X}_\epsilon)) + \text{err}} + \left( I - \text{diag}^* \left( \frac{\text{diag}(\widehat{X}_\epsilon) + \text{err}}{\max(\text{diag}(\widehat{X}_\epsilon)) + \text{err}} \right) \right)$$

The objective function value of (MA-Rel) at $X^f$ is

$$
\begin{aligned}
\langle C, X^f \rangle &= \left\langle C, \frac{\widehat{X}_\epsilon + \text{err}\mathbb{1}\mathbb{1}^T}{\max(\text{diag}(\widehat{X}_\epsilon)) + \text{err}} + \left( I - \text{diag}^* \left( \frac{\text{diag}(\widehat{X}_\epsilon) + \text{err}}{\max(\text{diag}(\widehat{X}_\epsilon)) + \text{err}} \right) \right) \right\rangle \\
&\underset{(i)}{\ge} \frac{\langle C, \widehat{X}_\epsilon \rangle}{\max(\text{diag}(\widehat{X}_\epsilon)) + \text{err}} + \left\langle C, \left( I - \text{diag}^* \left( \frac{\text{diag}(\widehat{X}_\epsilon) + \text{err}}{\max(\text{diag}(\widehat{X}_\epsilon)) + \text{err}} \right) \right) \right\rangle \\
&\underset{(ii)}{\ge} \frac{\langle C, \widehat{X}_\epsilon \rangle}{\max(\text{diag}(\widehat{X}_\epsilon)) + \text{err}} \underset{(iii)}{\ge} \frac{1 - 4\epsilon}{1 + 2\epsilon} \langle C, X_G^\star \rangle \underset{(iv)}{\ge} (1 - 6\epsilon)\langle C, X_G^\star \rangle \tag{27}
\end{aligned}
$$

where (i) follows from the fact that $\langle L_{G^-}, \text{err}\mathbb{1}\mathbb{1}^T \rangle = 0$ and $\langle W_+, \text{err}\mathbb{1}\mathbb{1}^T \rangle \ge 0$, (ii) follows since $L_{G^-}$ and $I - \text{diag}^* \left( \frac{\text{diag}(\widehat{X}_\epsilon) + \text{err}}{\max(\text{diag}(\widehat{X}_\epsilon)) + \text{err}} \right)$ are PSD and their inner product is nonnegative and the diagonal entries of $W_+$ are 0, (iii) follows from Lemma 5 and the fact that $\text{err} \le \epsilon$, and (iv) follows since $1 - 4\epsilon \ge (1 + 2\epsilon)(1 - 6\epsilon)$. Combining the fact that $\langle C, X_G^\star \rangle \ge \text{opt}_{CC}^G$ and $\mathbb{E}[\mathcal{C}] \ge 0.766\langle C, X^f \rangle$ with the above, we have

$$\mathbb{E}[\mathcal{C}] \ge 0.766(1 - 6\epsilon)\text{opt}_{CC}^G.$$

$\qquad\square$

## A.7 Proof of Lemma 7

*Proof.* We use Algorithm 1 with $p = \frac{\epsilon}{T(n,\epsilon)}$ and $T(n, \epsilon) = \frac{64 \log(2n + |E|)n^2}{\epsilon^2}$ to generate an $\epsilon\Delta$-optimal solution to (MA-LSE), where $\Delta = \text{Tr}(L_{G^-}) + \sum_{ij \in E^+} w_{ij}^+$.

**Upper bound on outer iteration complexity.** The convergence result given in Theorem 1 holds when Algorithm 1 is applied to (k-Cut-LSE). Then, the total number of iterations of Algorithm 1, also known as outer iteration complexity, required to generate $\epsilon\Delta$-optimal solution to (k-Cut-LSE) is

$$T = \frac{2C_g^u(1 + \eta)}{\epsilon\Delta} - 2 = \frac{2\beta M n^2 (1 + \eta)}{\epsilon\Delta} - 2.$$

**Bound on the value of generated clustering.** Algorithm 1 with $p = \frac{\epsilon}{T(n,\epsilon)}$ generates a solution $\widehat{X}_\epsilon$ that satisfies the bounds in Lemma 2 with probability with at least $1 - \epsilon$ after at most $T$ iterations. Thus, the bound given in (27) holds with probability at least $1 - \epsilon$ and we have

$$\mathbb{E}[\langle C, X^f \rangle] \geq (1 - 6\epsilon)\langle C, X_G^\star \rangle(1 - \epsilon) \geq (1 - 7\epsilon)\langle C, X_G^\star \rangle.$$

Let $\mathbb{E}_L[\cdot]$ denote the expectation over the randomness in the subproblem LMO and $\mathbb{E}_G[\cdot]$ denote the expectation over random Gaussian samples. The expected value of the generated clustering is then bounded as

$$\mathbb{E}[\mathcal{C}] = \mathbb{E}_L[\mathbb{E}_G[\mathcal{C}]] \underset{(i)}{\geq} 0.766\mathbb{E}_L[\langle C, X^f \rangle] \geq 0.766(1 - 7\epsilon)\langle C, X_G^\star \rangle \geq 0.766(1 - 7\epsilon)\mathrm{opt}_{CC}^G,$$

where (i) follows from the fact that the value of clustering generated by CGW rounding scheme satisfies $\mathbb{E}[\mathcal{C}] \geq 0.766\langle C, X^f \rangle$.

We now determine the inner iteration complexity of Algorithm 1.

**Upper bound on inner iteration complexity.** At each iteration $t$ of Algorithm 1, the subroutine LMO (see (18)) is equivalent to approximately computing maximum eigenvector of the matrix $\nabla g_t$. This is achieved using Lanczos method whose convergence is given in Lemma A.2. Now, let $N = \frac{1}{2} + \frac{1}{\rho}\log(n/p^2)$. We see that the bound on $1/\rho$ is

$$\frac{1}{\rho} \leq \frac{\alpha\|\nabla g_t\|(1 + \eta)}{4\eta\epsilon\Delta}, \tag{28}$$

which is similar to (20). We now derive an upper bound on $\|\nabla g_t\|$.

**Lemma A.5.** *Let $g(X) = \langle L_{G^-} + W^+, X \rangle - \beta\phi_M\left(\mathrm{diag}(X) - \mathbb{1}, \left[-e_i^T X e_j\right]_{(i,j)\in E}\right)$, where $\phi_M(\cdot)$ is defined in* (3). *We have $\|\nabla g_t\| \leq \Delta(1 + 4(\sqrt{2|E| + n}))$, where $\Delta = \mathrm{Tr}(L_{G^-}) + \sum_{ij\in E^+} w_{ij}^+$.*

*Proof.* For the function $g(X)$ as defined in the lemma, $\nabla g_t = L_{G^-} + W^+ - \beta D$, where $D$ is matrix such that $D_{ii} \in [-1, 1]$ for $i = 1, \ldots, n$, $D_{ij} \in [-1, 1]$ for $(i, j) \in E$, and $D_{ij} = 0$ for $(i, j) \notin E$, and $E = E^- \cup E^+$. We have

$$\max_k |\lambda_k(W^+)| \leq \sqrt{\mathrm{Tr}(W^{+T}W^+)} = \sqrt{\sum_{(i,j)\in E^+} |w_{ij}^+|^2} \leq \sum_{(i,j)\in E^+} w_{ij}^+, \quad \text{and} \tag{29}$$

$$\max_k |\lambda_k(D)| \leq \sqrt{\mathrm{Tr}(D^T D)} = \sqrt{\sum_{i,j=1}^n |D_{ij}|^2} \leq \sqrt{2|E| + n}, \tag{30}$$

where the last inequality follows since $D$ has at most $2|E| + n$ nonzero entries in the range $[-1, 1]$. Now, we have

$$\begin{aligned}
\|\nabla g_t\| = \|L_{G^-} + W^+ - \beta D\| &\underset{(i)}{\leq} \|L_{G^-}\| + \|W^+\| + \|-\beta D\| \\
&\leq \max_i |\lambda_i(L_{G^-})| + \max_i |\lambda_i(W^+)| + \max_i |\lambda_i(-\beta D)| \\
&\underset{(ii)}{\leq} \mathrm{Tr}(L_{G^-}) + \sum_{(i,j)\in E^+} w_{ij}^+ + \beta\sqrt{2|E| + n} \\
&\underset{(iii)}{\leq} \Delta(1 + 4(\sqrt{2|E| + n})).
\end{aligned}$$

where (i) follows since the spectral norm of $L_{G^-} + W^+ - \beta D$ satisfies the triangle inequality, (ii) follows from (29), (30) and the fact that $L_{G^-}$ is a positive semidefinite matrix, and (iii) follows by substituting the value of $\Delta$ and $\beta = 4\Delta$ as given in Lemma 5. $\qquad\square$

Substituting the bound on $\|\nabla g_t\|$ in (28), we have

$$\frac{1}{\rho} \leq \frac{1+\eta}{4\eta} \frac{n(1+4(\sqrt{2|E|+n}))}{\epsilon}, \quad \text{and}$$

$$N = \frac{1}{2} + \frac{1}{\sqrt{\rho}}\log(n/p^2) \leq \frac{1}{2} + \sqrt{\frac{1+\eta}{4\eta}}\sqrt{\frac{n(1+4(\sqrt{2|E|+n}))}{\epsilon}}\log(n/p^2) = N^u.$$

The computational complexity of Lanczos method is $\mathcal{O}(N^u(|E|+n))$, where the term $|E|+n$ appears since Lanczos method performs matrix-vector multiplication with $\|\nabla g_t\|$, whose sparsity is $\mathcal{O}(|E|)$, plus additional $\mathcal{O}(n)$ arithmetic operations at each iteration. We finally write the computational complexity of each iteration of Algorithm 1 as $\mathcal{O}\left(\frac{\sqrt{n}|E|^{1.25}}{\sqrt{\epsilon}}\log(n/p^2)\right)$.

**Total computational complexity of Algorithm 1.** Since $p = \frac{\epsilon}{T(n,\epsilon)}$, we have

$$\log\left(\frac{n}{p^2}\right) = \log\left(\frac{(64)^2 n^5(\log(2n+|E|))^2}{\epsilon^6}\right) \leq \log\left(\frac{4^6 n^6}{\epsilon^6}\right) = 6\log\left(\frac{4n}{\epsilon}\right),$$

where the inequality follows since $|E| \leq \binom{n-1}{2}$ and $\left(\log\left(2n+\binom{n-1}{2}\right)\right)^2 \leq n$ for $n \geq 1$. Multiplying outer and inner iteration complexity and substituting the bound on $p$, we prove that Algorithm 1 is a $\mathcal{O}\left(\frac{n^{2.5}|E|^{1.25}}{\epsilon^{2.5}}\log(n/\epsilon)\log(|E|)\right)$-time algorithm. $\square$

### A.8 Proof of Lemma 8

For any symmetric matrix $X \in \mathbb{S}^n$, the definition of $\tau$-spectral closeness (Definition 1) implies

$$(1-\tau)\langle L_G, X\rangle \leq \langle L_{\tilde{G}}, X\rangle \leq (1+\tau)\langle L_G, X\rangle. \tag{31}$$

Let $C$ and $\tilde{C}$ be the cost matrix in the objective of (k-Cut-Rel), when the problem is defined on the graphs $G$ and $\tilde{G}$ respectively. Since $C$ and $\tilde{C}$ are scaled Laplacian matrices (with the same scaling factor $(k-1)/2k$, from (31), we can write

$$(1-\tau)\langle C, X\rangle \leq \langle \tilde{C}, X\rangle \leq (1+\tau)\langle C, X\rangle. \tag{32}$$

Let $X_G^\star$ and $X_{\tilde{G}}^\star$ be optimal solutions to (k-Cut-Rel) defined on the graphs $G$ and $\tilde{G}$ respectively. From (32), we can write,

$$(1-\tau)\langle C, X_G^\star\rangle \leq \langle \tilde{C}, X_G^\star\rangle \leq \langle \tilde{C}, X_{\tilde{G}}^\star\rangle, \tag{33}$$

where the last inequality follows since $X_G^\star$ and $X_{\tilde{G}}^\star$ are feasible and optimal solutions respectively to (k-Cut-Rel) defined on the graph $\tilde{G}$. Combining this with the bound in Lemma 3, i.e., $\mathbb{E}[\texttt{CUT}] \geq \alpha_k(1-4\epsilon)\langle \tilde{C}, X_{\tilde{G}}^\star\rangle$, we get

$$\mathbb{E}[\texttt{CUT}] \geq \alpha_k(1-4\epsilon)\langle \tilde{C}, X_{\tilde{G}}^\star\rangle \underset{(i)}{\geq} \alpha_k(1-4\epsilon)(1-\tau)\langle C, X_G^\star\rangle \underset{(ii)}{\geq} \alpha_k(1-4\epsilon-\tau)\langle C, X_G^\star\rangle$$

$$\underset{(iii)}{\geq} \alpha_k(1-4\epsilon-\tau)\mathrm{opt}_k^G,$$

where (i) follows from (33), (ii) follows since $(1-4\epsilon)(1-\tau) = 1-4\epsilon-\tau+4\epsilon\tau \geq 1-4\epsilon\tau$ for nonnegative $\epsilon$ and $\tau$, and (iii) follows since $\langle C, X_G^\star\rangle \geq \mathrm{opt}_k^G$ for an optimal solution $X_G^\star$ to (k-Cut-Rel) defined on the graph $G$.

### A.9 Proof of Lemma 9

*Proof.* The Laplacian matrices $L_{G^-}$ and $L_{\tilde{G}^-}$ of the graphs $G^-$ and its sparse approximation $\tilde{G}^-$ respectively satisfy (31). Furthermore, let $L_G^+ = D^+ - W^+$, where $D_{ii}^+ = \sum_{j:(i,j)\in E^+} w_{ij}^+$, be the Laplacian of the graph $G^+$ and similarly let $L_{\tilde{G}}^+ = \tilde{D}^+ - \tilde{W}^+$ be the Laplacian of the graph $\tilde{G}^+$. If $X = I$, from (31), we have

$$(1-\tau)\mathrm{Tr}(D^+) \leq \mathrm{Tr}(\tilde{D}^+) \leq (1+\tau)\mathrm{Tr}(D^+). \tag{34}$$

Rewriting the second inequality in (31) for $X = X_G^\star$, and noting that $\text{diag}(X_G^\star) = \mathbb{1}$, we have

$$
\begin{aligned}
\langle W^+, X_G^\star \rangle &\leq \frac{\langle \tilde{W}^+, X_G^\star \rangle}{1+\tau} + \frac{(1+\tau)\text{Tr}(D^+) - \text{Tr}(\tilde{D}^+)}{1+\tau} \\
&\leq \frac{\langle \tilde{W}^+, X_G^\star \rangle}{1+\tau} + \frac{2\tau\text{Tr}(D^+)}{1+\tau},
\end{aligned}
\tag{35}
$$

where the second inequality follows from (34). Let $C = L_{G^-} + W^+$ and $\tilde{C} = L_{\tilde{G}^-} + \tilde{W}^+$ represent the cost in (MA-Rel) and (MA-Sparse) respectively. Let $X_G^\star$ be an optimal solution to (MA-Rel). The optimal objective function value of (MA-Rel) at $X_G^\star$ is $\langle C, X_G^\star \rangle$ and

$$
\begin{aligned}
(1-\tau)\langle C, X_G^\star \rangle &= (1-\tau)\langle L_{G^-}, X_G^\star \rangle + (1-\tau)\langle W^+, X_G^\star \rangle \\
&\underset{(i)}{\leq} \langle L_{\tilde{G}^-}, X_G^\star \rangle + \frac{1-\tau}{1+\tau}\langle \tilde{W}^+, X_G^\star \rangle + \frac{2\tau(1-\tau)}{1+\tau}\text{Tr}(D^+) \\
&\underset{(ii)}{\leq} \langle \tilde{C}, X_G^\star \rangle - \frac{2\tau}{1+\tau}\langle \tilde{W}^+, X_G^\star \rangle + \frac{2\tau}{1+\tau}\text{Tr}(\tilde{D}^+) \\
&\underset{(iii)}{\leq} \langle \tilde{C}, X_{\tilde{G}}^\star \rangle + \frac{2\tau}{1+\tau}\langle \tilde{C}, X_{\tilde{G}}^\star \rangle,
\end{aligned}
$$

where (i) follows from (31) and (35), (ii) follows from (34), and substituting $\tilde{C} = L_{\tilde{G}^-} + \tilde{W}^+$ and rearranging the terms and (iii) holds true since $\langle \tilde{W}^+, X_G^\star \rangle \geq 0$, and $I$ and $X_G^\star$ are feasible to (MA-Sparse) so that $\text{Tr}(\tilde{D}^+) \leq \langle \tilde{C}, X_{\tilde{G}}^\star \rangle$ and $\langle \tilde{C}, X_G^\star \rangle \leq \langle \tilde{C}, X_{\tilde{G}}^\star \rangle$. Rearraning the terms, we have

$$
\langle C, X_G^\star \rangle \leq \frac{1+3\tau}{1-\tau^2}\langle \tilde{C}, X_{\tilde{G}}^\star \rangle.
\tag{36}
$$

Combining (36) with the fact that the expected value of clustering $\mathbb{E}[\mathcal{C}]$ generated for the graph $\tilde{G}$ satisfies (11), we have

$$
\mathbb{E}[\mathcal{C}] \geq 0.766(1-6\epsilon)\langle \tilde{C}, X_{\tilde{G}}^\star \rangle \geq 0.766\frac{(1-6\epsilon)(1-\tau^2)}{1+3\tau}\langle C, X_G^\star \rangle \geq (1-6\epsilon-3\tau)(1-\tau^2)\text{opt}_{CC}^G,
$$

where the last inequality follows since $(1-6\epsilon-3\tau)(1+3\tau) \leq 1-6\epsilon$. $\qquad\square$

### A.10   Proof of Lemma 10

The first step of the procedure given in Section 5 is to sparsify the input graph using the technique proposed in [3] whose computational complexity is $\mathcal{O}(|E|\log^2 n)$. The second step when generating solutions to MAX-k-CUT and MAX-AGREE is to apply the procedures given in Sections 3 and 4 respectively. The computational complexity of this step is bounded as given in Propositions 4 and 7 leading to a $\mathcal{O}\left(\frac{n^{2.5}|E|^{1.25}}{\epsilon^{2.5}}\log(n/\epsilon)\log(|E|)\right)$-time algorithm.

**Bound on the value of generated $k$-cut.**   Let $p = \frac{\epsilon}{T(n,\epsilon)}$ and $T(n,\epsilon) = \frac{144\log(2n+|E|)n^2}{\epsilon^2}$ as given in Lemma 4. Using the procedure given in Section 3, we have $\mathbb{E}[\text{CUT}] \geq \alpha_k(1-5\epsilon)\text{opt}_k^{\tilde{G}}$. From the proof of Lemma 8, we see that CUT is then an approximate $k$-cut for the input graph $G$ such that $\mathbb{E}[\text{CUT}] \geq \alpha_k(1-5\epsilon-\tau)\text{opt}_k^G$.

**Bound on the value of generated clustering.**   Let $p = \frac{\epsilon}{T(n,\epsilon)}$ and $T(n,\epsilon) = \frac{64\log(2n+|E|)n^2}{\epsilon^2}$ as given in Lemma 7 and let the procedure given in Section 4 be applied to the sparse graph $\tilde{G}$. Then, the generated clustering satisfies $\mathbb{E}[\mathcal{C}] \geq 0.766(1-7\epsilon)\text{opt}_{CC}^{\tilde{G}}$. Combining this with the proof of Lemma 9, we have $\mathbb{E}[\mathcal{C}] \geq 0.766(1-7\epsilon-3\tau)(1-\tau^2)\text{opt}_{CC}^G$.

# B  Preliminary Computational Results for MAX-$k$-CUT

We provide some preliminary computational results when generating an approximate $k$-cut on the graph $G$ using the approach outlined in Section 3. The aim of these experiments was to verify that the bounds given in Lemma 3 were satisfied in practice. First, we solved (k-Cut-LSE) to $\epsilon\mathrm{Tr}(C)$-optimality using Algorithm 1 with the input parameters set to $\alpha = n$, $\epsilon = 0.05$, $\beta = 6\mathrm{Tr}(C)$, $M = 6\frac{\log(2n)+|E|}{\epsilon}$. We then computed feasible samples using Algorithm 2 and then finally used the FJ rounding scheme on the generated samples. The computations were performed using MATLAB R2021a on a machine with 8GB RAM. The peak memory requirement was noted using the `profiler` command in MATLAB.

We performed computations on randomly selected graphs from GSET dataset. In each case, the infeasibility of the covariance of the generated samples was less than $\epsilon$, thus satisfying (7). The number of iterations of LMO in Algorithm 1 was also within the bounds given in Proposition 1. To a generate $k$-cut, we generated 10 sets of $k$ i.i.d. zero-mean Gaussian samples with covariance $\widehat{X}_\epsilon$, and each set was then used to generate a $k$-cut for the input graph using FJ rounding scheme. Let $\mathrm{CUT}_{\mathrm{best}}$ denote the value of the best $k$-cut amongst the 10 generated cuts. Table 1 shows the result for graphs from the GSet dataset with $k = 3, 4$. Note that, $\mathrm{CUT}_{\mathrm{best}} \geq \mathbb{E}[\mathrm{CUT}] \geq \alpha_k(1 - 4\epsilon)\langle C, X^\star\rangle \geq \alpha_k\frac{1-4\epsilon}{1+4\epsilon}\langle C, \widehat{X}_\epsilon\rangle$, where the last inequality follows from combining (8) with (6). Since we were able to generate the values, $\mathrm{CUT}_{\mathrm{best}}$ and $\langle C, \widehat{X}_\epsilon\rangle$, we noted that the weaker bound $\mathrm{CUT}_{\mathrm{best}}/\langle C, \widehat{X}_\epsilon\rangle = \mathrm{AR} \geq \alpha_k(1 - 4\epsilon)/(1 + 4\epsilon)$ was satisfied by every input graph when $\epsilon = 0.05$.

Furthermore, Table 1 also shows that the memory used by our method was linear in the size of the input graph. To see this, consider the dataset G1, and note that for $k = 3$, the memory used by our method was $1252.73\mathrm{kB} \approx 8.02 \times (|V| + |E|) \times 8$, where a factor of 8 denotes that MATLAB uses 8 bytes to store a real number. Similarly, for other instances in GSET, the memory used by our method to generate an approximate $k$-cut for $k = 3, 4$ was at most $c \times (|V| + |E|) \times 8$, where for each graph the value of $c$ was bounded by $c \leq 82$, showing linear dependence of the memory used on the size of the input graph.

Table 1: Result of generating a $k$-cut for graphs from GSET using the method outlined in Section 3. We have, infeas $= \max\{\|\mathrm{diag}(X) - 1\|_\infty, \max\{0, -[\widehat{X}_\epsilon]_{ij} - \frac{1}{k-1}\}\}$ and AR $= \mathrm{CUT}_{\mathrm{best}}/\langle C, \widehat{X}_\epsilon\rangle$.

| Dataset | $|V|$ | $|E|$ | $k$ | # Iterations $(\times 10^3)$ | infeas | $\langle C, \widehat{X}_\epsilon\rangle$ | $\mathrm{CUT}_{\mathrm{best}}$ | AR | Memory required (in kB) |
|---|---|---|---|---|---|---|---|---|---|
| G1 | 800 | 19176 | 3 | 823.94 | $4 \times 10^{-4}$ | 15631 | 14266 | 0.9127 | 1252.73 |
| G1 | 800 | 19176 | 4 | 891.23 | $4 \times 10^{-4}$ | 17479 | 15746 | 0.9 | 1228.09 |
| G2 | 800 | 19176 | 3 | 827.61 | $6 \times 10^{-5}$ | 15629 | 14332 | 0.917 | 1243.31 |
| G2 | 800 | 19176 | 4 | 9268.42 | $8 \times 10^{-5}$ | 17474 | 15786 | 0.903 | 1231.07 |
| G3 | 800 | 19176 | 3 | 1242.53 | $7 \times 10^{-5}$ | 15493 | 14912 | 0.916 | 1239.57 |
| G3 | 800 | 19176 | 4 | 1341.37 | $7 \times 10^{-45}$ | 17301 | 15719 | 0.908 | 1240.17 |
| G4 | 800 | 19176 | 3 | 812.8 | $9 \times 10^{-5}$ | 15660 | 14227 | 0.908 | 1230.59 |
| G4 | 800 | 19176 | 4 | 9082.74 | $10^{-4}$ | 17505 | 15748 | 0.899 | 1223.59 |
| G5 | 800 | 19176 | 3 | 843.5 | $10^{-4}$ | 15633 | 14341 | 0.917 | 1222.09 |
| G5 | 800 | 19176 | 4 | 9294.32 | $10^{-4}$ | 17470 | 15649 | 0.895 | 1227.9 |
| G14 | 800 | 4694 | 3 | 1240.99 | 0.002 | 3917 | 2533 | 0.646 | 3502.64 |
| G14 | 800 | 4694 | 4 | 3238.42 | 0.001 | 4467.9 | 3775 | 0.844 | 519.25 |
| G15 | 800 | 4661 | 3 | 3400.17 | 0.001 | 4018.6 | 3385 | 0.842 | 612 |
| G15 | 800 | 4661 | 4 | 1603.13 | 0.001 | 4475.8 | 3754 | 0.838 | 648.17 |
| G16 | 800 | 4672 | 3 | 33216.68 | 0.001 | 4035.7 | 3422 | 0.847 | 561 |
| G16 | 800 | 4672 | 4 | 3059.11 | 0.001 | 4437.5 | 3783 | 0.852 | 2800 |
| G17 | 800 | 4667 | 3 | 3526.4 | 0.001 | 4031.5 | 3414 | 0.846 | 602.81 |
| G17 | 800 | 4667 | 4 | 3400.01 | 0.001 | 4440 | 3733 | 0.84 | 693.6 |
| G22 | 2000 | 19990 | 3 | 7402.59 | $10^{-4}$ | 17840 | 11954 | 0.67 | 1340.34 |
| G22 | 2000 | 19990 | 4 | 8103.83 | $10^{-4}$ | 19582 | 16670 | 0.851 | 1341.67 |
| G23 | 2000 | 19990 | 3 | 3597.39 | $10^{-4}$ | 17938 | 15331 | 0.854 | 1360.09 |
| G23 | 2000 | 19990 | 4 | 3588.04 | $10^{-4}$ | 19697 | 16639 | 0.844 | 1317.09 |
| G24 | 2000 | 19990 | 3 | 4304.48 | $10^{-4}$ | 17913 | 15370 | 0.858 | 1341.96 |
| G24 | 2000 | 19990 | 4 | 1994.26 | $10^{-4}$ | 19738 | 16624 | 0.842 | 1321.59 |

*Continued on next page*

| Dataset | $|V|$ | $|E|$ | $k$ | # Iterations $(\times 10^3)$ | infeas | $\langle C, \widehat{X}_\epsilon \rangle$ | CUT$_{\text{best}}$ | AR | Memory required (in kB) |
|---|---|---|---|---|---|---|---|---|---|
| G25 | 2000 | 19990 | 3 | 9774.03 | $10^{-4}$ | 18186 | 15294 | 0.841 | 1311.54 |
| G25 | 2000 | 19990 | 4 | 1540.14 | $10^{-4}$ | 19778 | 16641 | 0.841 | 1330.95 |
| G26 | 2000 | 19990 | 3 | 2069.65 | $10^{-4}$ | 18012 | 15411 | 0.855 | 1321.92 |
| G26 | 2000 | 19990 | 4 | 1841.06 | $2 \times 10^{-4}$ | 19735 | 16609 | 0.841 | 1331.53 |
| G43 | 1000 | 9990 | 3 | 894.53 | $10^{-4}$ | 9029 | 7785 | 0.862 | 661.09 |
| G43 | 1000 | 9990 | 4 | 9709.68 | $2 \times 10^{-4}$ | 9925 | 8463 | 0.852 | 665.59 |
| G44 | 1000 | 9990 | 3 | 721.64 | $10^{-4}$ | 9059.5 | 7782 | 0.859 | 661.09 |
| G44 | 1000 | 9990 | 4 | 9294.43 | $10^{-4}$ | 9926.1 | 8448 | 0.851 | 765.37 |
| G45 | 1000 | 9990 | 3 | 794.84 | $10^{-4}$ | 9038.4 | 7773 | 0.86 | 661.09 |
| G45 | 1000 | 9990 | 4 | 9503.74 | $2 \times 10^{-4}$ | 9929.7 | 8397 | 0.845 | 669 |
| G46 | 1000 | 9990 | 3 | 703.4 | $10^{-4}$ | 9068.5 | 7822 | 0.862 | 661.09 |
| G46 | 1000 | 9990 | 4 | 9684.93 | $4 \times 10^{-4}$ | 9929.9 | 8333 | 0.839 | 657.09 |
| G47 | 1000 | 9990 | 3 | 777.61 | $10^{-4}$ | 9059.4 | 7825 | 0.863 | 679.89 |
| G47 | 1000 | 9990 | 4 | 9789.55 | $2 \times 10^{-4}$ | 9930.8 | 8466 | 0.852 | 661.09 |

## C   Additional Computational Results for Correlation Clustering

We provide the computational result for the graphs from the GSET dataset (not included in the main article) here. We performed computations for graphs G1-G57 from GSET dataset. The instances for which we were able to generate an $\epsilon\Delta$-optimal solution to (MA-LSE) are given in Table 2, where the parameters, $\epsilon$ and $\Delta$, were set as given in Section 6. For the instances not in the table, we were not able to generate an $\epsilon\Delta$-optimal solution after 30 hours of runtime.

Table 2: Result of generating a clustering of graphs from GSET using the method outlined in Section 4. We have, infeas $= \max\{\|\text{diag}(X) - 1\|_\infty, \max\{0, -[\widehat{X}_\epsilon]_{ij}\}\}$, AR $= \mathcal{C}_{\text{best}}/\langle C, \widehat{X}_\epsilon \rangle$ and $0.75(1 - 6\epsilon)/(1 + 4\epsilon) = 0.4375$ for $\epsilon = 0.05$.

| Dataset | $|V|$ | $|E^+|$ | $|E^-|$ | # Iterations $(\times 10^3)$ | infeas | $\langle C, \widehat{X}_\epsilon \rangle$ | $\mathcal{C}_{\text{best}}$ | AR | Memory required (in kB) |
|---|---|---|---|---|---|---|---|---|---|
| G2 | 800 | 2501 | 16576 | 681.65 | $8 \times 10^{-4}$ | 848.92 | 643.13 | 0.757 | 1520.18 |
| G3 | 800 | 2571 | 16498 | 677.56 | $7 \times 10^{-4}$ | 835.05 | 634.83 | 0.76 | 1529.59 |
| G4 | 800 | 2457 | 16622 | 665.93 | $6 \times 10^{-4}$ | 852.18 | 647.37 | 0.76 | 1752 |
| G5 | 800 | 2450 | 16623 | 646.4 | $10^{-3}$ | 840.63 | 636.21 | 0.756 | 1535.92 |
| G6 | 800 | 9665 | 9511 | 429.9 | $3 \times 10^{-4}$ | 25766 | 21302 | 0.826 | 1664 |
| G7 | 800 | 9513 | 9663 | 423.58 | $8 \times 10^{-4}$ | 26001 | 20790 | 0.799 | 1535.06 |
| G8 | 800 | 9503 | 9673 | 421.34 | $6 \times 10^{-4}$ | 26005 | 21080 | 0.81 | 4284 |
| G9 | 800 | 9556 | 9620 | 426.4 | $3 \times 10^{-4}$ | 25903 | 21326 | 0.823 | 1812 |
| G10 | 800 | 9508 | 9668 | 426.25 | $3 \times 10^{-4}$ | 25974 | 21412 | 0.824 | 1535.59 |
| G12 | 800 | 798 | 802 | 393.69 | $9 \times 10^{-4}$ | 3023.4 | 2034 | 0.672 | 444.06 |
| G13 | 800 | 817 | 783 | 416.29 | $8 \times 10^{-4}$ | 3001.1 | 2010 | 0.669 | 613.03 |
| G15 | 800 | 3801 | 825 | 284.77 | $10^{-3}$ | 529.83 | 401.19 | 0.757 | 460.17 |
| G16 | 800 | 3886 | 749 | 228.12 | $8 \times 10^{-4}$ | 524.69 | 417.88 | 0.796 | 451.07 |
| G17 | 800 | 3899 | 744 | 2448.633 | $9 \times 10^{-4}$ | 536.65 | 369.04 | 0.687 | 480.45 |
| G18 | 800 | 2379 | 2315 | 1919.44 | $2 \times 10^{-3}$ | 7237.6 | 5074 | 0.701 | 434.67 |
| G19 | 800 | 2274 | 2387 | 2653.79 | $2 \times 10^{-3}$ | 7274.2 | 5130 | 0.705 | 496 |
| G20 | 800 | 2313 | 2359 | 1881.75 | $2 \times 10^{-3}$ | 7258.1 | 5186 | 0.714 | 406.09 |
| G21 | 800 | 2300 | 2367 | 1884.97 | $2 \times 10^{-3}$ | 7281.3 | 5238 | 0.719 | 467.26 |
| G23 | 2000 | 120 | 19855 | 550.77 | $2 \times 10^{-3}$ | 1802.2 | 1373.2 | 0.762 | 1651.54 |
| G24 | 2000 | 96 | 19875 | 812.16 | $10^{-3}$ | 1811.2 | 1384.6 | 0.764 | 1678.04 |
| G25 | 2000 | 109 | 19872 | 1739.06 | $6 \times 10^{-4}$ | 1801.8 | 1398.1 | 0.776 | 1650.48 |
| G26 | 2000 | 117 | 19855 | 1125.74 | $10^{-3}$ | 1789.9 | 1356.9 | 0.758 | 1650.01 |
| G27 | 2000 | 9974 | 10016 | 464.93 | $5 \times 10^{-4}$ | 30502 | 22010 | 0.721 | 1647.09 |

*Continued on next page*

Table 2 – *Continued from previous page*

| Dataset | $|V|$ | $|E^+|$ | $|E^-|$ | # Iterations ($\times 10^3$) | infeas | $\langle C, \widehat{X}_\epsilon \rangle$ | $\mathcal{C}_{\text{best}}$ | AR | Memory required (in kB) |
|---|---|---|---|---|---|---|---|---|---|
| G28 | 2000 | 9943 | 10047 | 553.65 | $4 \times 10^{-4}$ | 30412 | 22196 | 0.729 | 1317.78 |
| G29 | 2000 | 10035 | 9955 | 513.97 | $2 \times 10^{-4}$ | 30366 | 23060 | 0.759 | 1310.46 |
| G30 | 2000 | 10045 | 9945 | 594.09 | $3 \times 10^{-4}$ | 30255 | 22550 | 0.745 | 1310.48 |
| G31 | 2000 | 9955 | 10035 | 1036.9 | $2 \times 10^{-4}$ | 29965 | 22808 | 0.761 | 1305.05 |
| G33 | 2000 | 1985 | 2015 | 403.75 | $10^{-3}$ | 7442 | 4404 | 0.591 | 634.93 |
| G34 | 2000 | 1976 | 2024 | 863.53 | $4 \times 10^{-4}$ | 7307.2 | 4760 | 0.651 | 574.12 |
| G44 | 1000 | 229 | 9721 | 515.18 | $10^{-3}$ | 810.82 | 616.61 | 0.76 | 655.09 |
| G45 | 1000 | 218 | 9740 | 504.91 | $10^{-3}$ | 812.21 | 615.84 | 0.758 | 660.51 |
| G46 | 1000 | 237 | 9723 | 469.6 | $10^{-3}$ | 818.39 | 623.95 | 0.762 | 655.09 |
| G47 | 1000 | 230 | 9732 | 495.24 | $9 \times 10^{-4}$ | 819.63 | 621.65 | 0.758 | 648.32 |
| G49 | 3000 | 0 | 6000 | 1002.59 | 0.003 | 599.64 | 456.48 | 0.761 | 733 |
| G50 | 3000 | 0 | 6000 | 996.19 | 0.004 | 599.64 | 455.78 | 0.76 | 540.26 |
| G52 | 1000 | 4750 | 1127 | 2041.8 | 0.001 | 684.1 | 441.02 | 0.644 | 757.59 |
| G53 | 1000 | 4820 | 1061 | 785.33 | $8 \times 10^{-4}$ | 695.53 | 445.03 | 0.639 | 417.07 |
| G54 | 1000 | 4795 | 1101 | 2899.99 | $7 \times 10^{-4}$ | 686.8 | 482.57 | 0.702 | 517.09 |
| G56 | 5000 | 6222 | 6276 | 1340.35 | 0.004 | 22246 | 12788 | 0.574 | 1243.98 |