# OpenReview forum: "Memory-Efficient Approximation Algorithms for Max-k-Cut and Correlation Clustering"
_NeurIPS.cc/2021/Conference — NeurIPS 2021 Poster_

### Official Review · Reviewer_qqhg · 2021-07-14

**Rating:** 4
**Confidence:** 2

**Summary:**

The authors study two graph partitioning problems MAX-K-CUT and correlation clustering (MAX-AGREE) under a limited memory setting. The authors propose polytime sampling-based algorithms for these two problems that require $O(n + |E|)$ memory while preserving the best approximation guarantees ($n$ is the number of vertices and $E$ is the edge set). In addition, for the case of dense graphs the authors eliminate the quadratic memory requirements by utilizing spectral sparsification results in the streaming model of computation.



**Limitations And Societal Impact:**

Yes, they did.

**Main Review:**

* In my opinion, the manuscript could be a better fit to an algorithmic / graph theory related conference rather than NeurIPS. Graph sparsification and approximation algorithms for graph related problems have mostly appeared in such conferences (SODA,ICALP, etc).
* What is the contribution of the current paper following [27]? Please elaborate on the contributions section how you "...modify their approach..." in L49-L50 at this point. This is your key contribution and should be clearly explained in this level of abstraction.
* Memory reduction using graph sparsification is a direct outcome of the spectral sparsification result.
* Correlation clustering / Literature Review: Is the result of Ahn et al [1] superior compared to your proposed algorithm? Memory requirements in [1] is $\tilde{O}(n/\epsilon^2)$, right? Moreover, [1] is a single pass algorithm. Please clarify.
* Experimental results are based on relatively small graphs 1-2MBs memory footprint. Here, the reader would expect instances of large graphs, at least sparse once to see the benefits of the proposed methods.
* Notation: Please do not hide logarithmic terms on n using $\tilde{O}$. Such notations can be used to hide $loglogn$ terms, etc.


*******************
Minor comments
*******************


Abstract:
* L8: E is undefined (edge set).


Introduction
* L33: "Equation (3)" is a forward reference (next 2 pages). Hard for the reader to follow the flow.
* L35: "for $l\in {1, \dots , k}$" . Is this "for all" or "for every"?
* L39: Equation on second \sum should read "ij\inE^{+}", not E^{-}.
* L40: Give reference to the SDP relaxation of MAX-AGREE or state if it is folklore.
* L45-46: "...uses them to define up to $8$ clusters". The 3 Gaussian samples here and 8 clusters seem arbitrary here. Please elaborate or add a sentence commenting about these numbers.
* L48: It would be great to add a sentence in the "1.3 Contribution Section". Enumerating the contributions directly is quite dense.
* L50: "nearly the same". Please be precise if possible here.
* L53: "$1-4\epsilon$". $\epsilon$ is not defined here, only on the proposition below.
* L69: "arrives edge-by-edge": Is the streaming model adversary or not? I am not an expert but it is good to make it clear to the reader.






**Time Spent Reviewing:**

4

---

> ### Author Response · Authors · 2021-08-10
> **Comparison with [1] and [27], and responses to questions by reviewer**
>
> We would like to thank the reviewer for the detailed feedback. We have added the responses to the comments below.
>
> - What is the contribution of the current paper following [27]? Please elaborate on the contributions section how you "...modify their approach..." in L49-L50 at this point. This is your key contribution and should be clearly explained in this level of abstraction. **Answer:** We exploit particular special properties of the rounding schemes and their analysis for Max-k-Cut and Max-Agree, which allows us to keep only $|E|$ of these constraints while maintaining the approximation quality of the solution to these problems. This allows us to extend the application of the Gaussian sampling-based Frank-Wolfe technique proposed in [27] (primarily for SDPs with $|V|$ number of constraints), the two problems that have $|V|^2$ number of constraints. Furthermore, we use rounding techniques tailored to Max-k-Cut and Max-Agree to generate zero-mean Gaussian samples with covariance equal to an approximate solution to the input problem. We also show that this method can be combined with spectral sparsification to generate a solution with provable optimality guarantees to problems defined on dense input graphs.
>
> - Correlation clustering / Literature Review: Is the result of Ahn et al [1] superior compared to your proposed algorithm? Memory requirements in [1] is $\tilde{\mathcal{O}}(n/\epsilon^2)$, right? Moreover, [1] is a single pass algorithm. Please clarify. **Answer:** Our memory usage is $\mathcal{O}(n+\min\{ |E|, \frac{n\log n}{\epsilon^2}\})$ which is the same as [1] for dense graphs but improves upon [1] for sparse graphs. Moreover, our memory usage is independent of $\epsilon$ for sparse graphs. In terms of running time, our algorithm improves on that in [1] only in terms of its dependence on $\epsilon$ (it is worse in terms of dependence on $|V|$ and $|E|$). Our method has $\mathcal{O}(\epsilon^{-2})$ time complexity in the sparse case, and $\mathcal{O}(\epsilon^{-4})$ in the dense case. While the algorithm of Ahn et al [1] has time complexity $\mathcal{O}(\epsilon^{-10})$ by comparison.
>
> - Experimental results are based on relatively small graphs 1-2MBs memory footprint. Here, the reader would expect instances of large graphs, at least sparse once to see the benefits of the proposed methods. **Answer:** Currently, our method is too slow to be practically used for large-scale instances. Numerical experiments for small-scale problems validate our theoretical results and demonstrate that the algorithm is feasible to implement. We note that the simplicity of the algorithm leaves open the possibility for significant algorithmic engineering to potentially improve the run-time.
>
> - Notation: Please do not hide logarithmic terms on n using $\tilde{\mathcal{O}}$. Such notations can be used to hide $\log\log n$ terms, etc. **Answer:** The computational complexity on L107 ($\tilde{\mathcal{O}}(n^3/\epsilon^4)$) is $\mathcal{O}(n^3\log  n\log (|E|+2n)/\epsilon^4)$. We will explicitly state our results in the revised manuscript if the paper is accepted.
>
> - L45-46: "...uses them to define up to 8 clusters". The 3 Gaussian samples here and 8 clusters seem arbitrary here. Please elaborate or add a sentence commenting about these numbers. **Answer:** The rounding scheme discussed on L45-46 is adapted from [10], and proposes to generate either $i=2$ or $i=3$ samples and use these samples to generate $2^i$ clusters, which is at most 8.

---

### Official Review · Reviewer_6muF · 2021-07-16

**Rating:** 6
**Confidence:** 4

**Summary:**

The methods with best approximation guarantees for max-K-cut and the max-agree variant  of correlation clustering involve solving semidefinte programs (SDPs) with $n^2$ variables and $n^2$ constraints, which use  a lot of memory. 	This paper proposes a modified polynomial-time Gaussian sampling-based algorithms from [27] that reduce the memory from $\mathcal{O}(n^2)$ to $\mathcal{O}(n + |E|)$ and nearly achieve the best approximation guarantees. The paper further extend their approach to the dense graphs cases by combining it with graph sparsification and the memory is reduced to $\mathcal{O}(nlog n / \tau^2)$.

**Limitations And Societal Impact:**

Yes

**Main Review:**


Originality & presentation:


- The idea of the paper is based on the observation that the approximation guarantee depends only on $X_{i,j}, (i, j) \in E$. Therefore, they define new SDP relaxations of Max-K-Cut and Max-Agree with $n+|E|$ constraints. The Frank-Wolfe algorithm with Gaussian sampling is then used to obtain the $\epsilon$-optimal solutions of these two relaxations. The algorithms only use $\mathcal{O}(n + |E|)$ memory since the relaxations have only $n+|E|$ constraints. For dense graphs, the memory will still be high and the paper tackles this issue by combining their approach with spectral graph sparsification and the memory is reudced to $\mathcal{O}(nlog n / \tau^2)$. The proposed approach nearly obtains the same quality solutions as previous work but uses less memory.

- The idea of the paper is intriguing and the reduction of the memory used by the algorithms is impressive. However, the computational complexity of the presented approach is in $\mathcal{O}(n^2|E|log(2n + |E|)/\epsilon^2)$, which makes it hard to be used in large-scale instances.


- One of techniques of the paper,  Gaussian sampling-based Frank-Wolfe, was introdudced in [27]. Can you explain the major differences or overlaps compared to [27]?

- What is the convergence  rate of the algorithm for MA-SDPs in Section 4?


Overall, I like the theoretical contribution of the paper.

The paper is well structured and main techniques are clearly presented.



Experiments:


In the experiments, the paper presents only the computational results of their own algorithm. There is not clear that it is able to outperform the previous baselines in practical.


**Time Spent Reviewing:**

6

---

> ### Author Response · Authors · 2021-08-10
> **Comparison with [27] and responses to questions by reviewer**
>
> We would like to thank the reviewer for the feedback. We have added responses to the specific queries raised.
>
> - One of techniques of the paper, Gaussian sampling-based Frank-Wolfe, was introdudced in [27]. Can you explain the major differences or overlaps compared to [27]? **Answer:**  The Gaussian sampling-based Frank-Wolfe technique in [27] focuses on problems with $n$ linear constraints, where $n$ is size of the decision variable $X\in \mathbb{S}^n_+$. We have extended their approach to Max-k-Cut and the Max-Agree version of correlation clustering, each of which has $n^2$ linear (inequality) constraints. To do this we exploit particular special properties of the rounding schemes and their analysis for these problems that allows us to keep only $|E|$ of these constraints without compromising approximation quality. Furthermore, we use rounding techniques tailored to Max-k-Cut and Max-Agree to generate zero-mean Gaussian samples with covariance equal to an approximate solution to the input problem.
>
> - What is the convergence rate of the algorithm for MA-SDPs in Section 4? **Answer:** We have a $\mathcal{O}(n^2|E|\log (2n+|E|)/\epsilon^2)$-time algorithm to generate an $0.766(1-6\epsilon)$-optimal solution to Max-Agree.

---

### Official Review · Reviewer_kFS9 · 2021-07-17

**Rating:** 7
**Confidence:** 3

**Summary:**

This paper studies several methods for speeding up / reducing memory of computing max k-way cuts, via methods such as sampling and sparsification. They rigorously prove that the semi-definite program can be approximated to an error of \epsilon in about m \epsilon^{-2} space, and furthermore, the quality of approximations are preserved with the use of spectral graph sparsifiers.

A Matlab implemention of this method is then tested on a multitude of clustering instances, showing that the memory usage is quite low, but at a cost of a fairly large (between 15~40%) relative error.

**Limitations And Societal Impact:**

I'm most concerned about the experiments: the scale of the data (5000 vertices, 20000 edges) is still at a scale where the entire semidefinite program fits into 100MB or so of memory, and can be solved as such. On the other hand, iterative / first order methods for solving semidefinite programs tend to converge significantly slower. So my interpretation of the data is that it's getting a 100 fold memory reduction at the cost of a 10^3~10^4 fold increase in runtime (10^6 iterations, vs the rougly factor 100 overhead of matrix inversion vs. multiplication), as well as a fairly large loss in accuracy. So I view the experiments mostly demonstrate that these methods can feasibly be implemented, instead of showing that they have clear gains over existing optimization tools.

**Main Review:**

I believe the bounds shown in this paper are quite useful for (approximately) solving semi-definite programs, especially in settings with limited memory. On the other hand, I feel the style of algorithm, and tools used, place this result one step away from being truly practical. My understanding is that clustering / inference on graphs tend to use more features of the datasets, instead of just the underlying graph structure. While I believe many of the ideas used here can be incorporated into more sophisticated semidefinite programming based algorithms, I feel the direct application to k-cut will likely be of interest primarily to those working on continuous optimization and streaming algorithms.

I also thank the authors for their responses that agree about this result being an intermediate result. My overall score is unchanged because I believe there are enough ideas in this result for it to be of interest to many.

**Time Spent Reviewing:**

1

---

> ### Author Response · Authors · 2021-08-10
> **Implementability of our algorithm**
>
> We would like to thank the reviewer for the encouraging feedback. We agree that the numerical experiments in the paper are focused on demonstrating the feasibility of the algorithm. However, we are a step closer to practical memory-efficient algorithms for Max-k-Cut and Max-Agree version of correlation clustering. Previous algorithms that are both memory efficient and have strong approximation guarantees (see, for e.g., [1]), have not, to our knowledge, been implemented.

---

### Official Review · Reviewer_xu1n · 2021-07-20

**Rating:** 7
**Confidence:** 2

**Summary:**

The paper presents an algorithm for approximating MAX-$k$-CUT to within the SDP bound that avoids the $\Omega(n^2)$ memory requirement of the SDP formulation. The improvement is to $O(n+m)$ space. Here, $n$ is the number of nodes of the input graph and $m$ is the number of edges. Coupled with known sparsification techniques, the algorithm runs using $O(n\log n)$ space (hiding a factor that depends on how close one gets to the SDP guarantee. The algorithm uses Frank-Wolfe with Gaussian sampling on a relaxation that replaces the constraints by some regularization. Similar results apply to correlation clustering, though such bounds were known previously.

**Ethical Concerns:**

None.

**Limitations And Societal Impact:**

None.

**Main Review:**

It's a nice result which uses mostly previously known techniques, but the application requires some effort. The problems are justified in the machine learning context as clustering problems, and the practical motivation for reducing substantially the memory requirements is indeed meaningful. There are some empirical results on randomly generated graphs. The paper has some annoying typos, for instance in line 40 and the preceding displayed equation.

**Time Spent Reviewing:**

4 hours

---

> ### Author Response · Authors · 2021-08-10
> **Comparison of our work with Ahn et al. [1]**
>
> We would like to thank the reviewer for their helpful feedback. We would like to briefly comment on how our results compare with Ahn et al [1]. Our memory usage is $\mathcal{O}(n+\min\{ |E|, \frac{n\log n}{\epsilon^2}\})$ which is the same as [1] for dense graphs but improves upon [1] for sparse graphs. Moreover, our memory usage is independent of $\epsilon$ for sparse graphs. In terms of running time, our algorithm improves on that in [1] only in terms of its dependence on $\epsilon$ (it is worse in terms of dependence on $|V|$ and $|E|$). Our method has $\mathcal{O}(\epsilon^{-2})$ time complexity in the sparse case, and $\mathcal{O}(\epsilon^{-4})$ in the dense case. While the algorithm of Ahn et al [1] has time complexity $\mathcal{O}(\epsilon^{-10})$ by comparison.

---

### Decision · Program_Chairs · 2021-09-27

**Decision:**

Accept (Poster)

**Comment:**

This submission was universally appreciated for its application to SDP-based methods for clustering but with lower memory requirements. Though, there were some suggestions that the paper might be more of a natural fit at a TCS conference instead of NeurIPS.